# ROBUST BACKDOOR ATTACK WITH VISIBLE, SEMANTIC, SAMPLE-SPECIFIC AND COMPATIBLE TRIGGERS

## ABSTRACT

Deep neural networks (DNNs) can be manipulated to exhibit specific behaviors when exposed to specific trigger patterns, without affecting their performance on benign samples, dubbed *backdoor attack*. Some recent research has focused on designing invisible triggers for backdoor attacks to ensure visual stealthiness, while showing high effectiveness, even under backdoor defense. However, we find that these carefully designed invisible triggers are often sensitive to visual distortion during inference, such as Gaussian blurring or environmental variations in physical scenarios. This phenomenon could significantly undermine the practical effectiveness of attacks, but has been rarely paid attention and thoroughly investigated. To address this limitation, we define a novel trigger called the **V**isible, **S**emantic, **S**ample-Specific, and **C**ompatible trigger (**VSSC** trigger), to achieve effective, stealthy and robust to visual distortion simultaneously. To implement it, we develop an innovative approach by utilizing the powerful capabilities of large language models for choosing the suitable trigger and text-guided image editing techniques for generating the poisoned image with the trigger. Extensive experimental results and analysis validate the effectiveness, stealthiness and robustness of the VSSC trigger. It demonstrates superior robustness to distortions compared with most of digital backdoor attacks and allows more efficient and flexible trigger integration compared to physical backdoor attacks. We hope that the proposed VSSC trigger and implementation approach could inspire future studies on designing more practical triggers in backdoor attacks.

## 1 INTRODUCTION

Deep neural networks (DNNs) have been successfully adopted in a wide range of important fields, such as face recognition (Balaban, 2015), verbal identification (Boles & Rad, 2017), object classification and detection (Zhao et al., 2019), and autonomous vehicles (Schwarting et al., 2018). However, DNNs face numerous security threats. One typical threat is backdoor attack, which can make DNNs perform specific behaviors when encountering a **particular trigger pattern** without affecting the performance on benign samples. This goal can be achieved by manipulating the training dataset or controlling the training process. In this work, we focus on the former threat model, *i.e.*, **data poisoning based backdoor attack**, and especially against the image classification task.

In the literature, several seminal works have been developed to ensure that the designed backdoor trigger is stealthy, effective, and resistant to backdoor defense. According to trigger visibility with respect to human visual perception, existing triggers can be categorized into visible and invisible triggers. Some early backdoor attacks adopted visible triggers (*e.g.*, BadNets (Gu et al., 2017) and TrojanNN (Liu et al., 2018b)) and showed a high attack success rate. However, it is easy to arouse human suspicion about visible triggers. Thus, recent works tend to design invisible triggers via image stenography (*e.g.*, SSBA (Li et al., 2021d) or slight spatial transformation (*e.g.*, WaNet (Nguyen & Tran, 2021)). And with some other characteristics (*e.g.*, sample-specific), these triggers showed their effectiveness even under several backdoor defenses. However, we experimentally find that the small magnitude of these invisible triggers cause the sensitivity to visual distortions, which may happen on each individual image at the inference stage. As shown in Figure 1, when conducting a common image processing like Gaussian Blur on the testing poisoned image (see the second row), or printing the testing poisoned image onto the paper in the physical scenario (see the bottom row), the attack

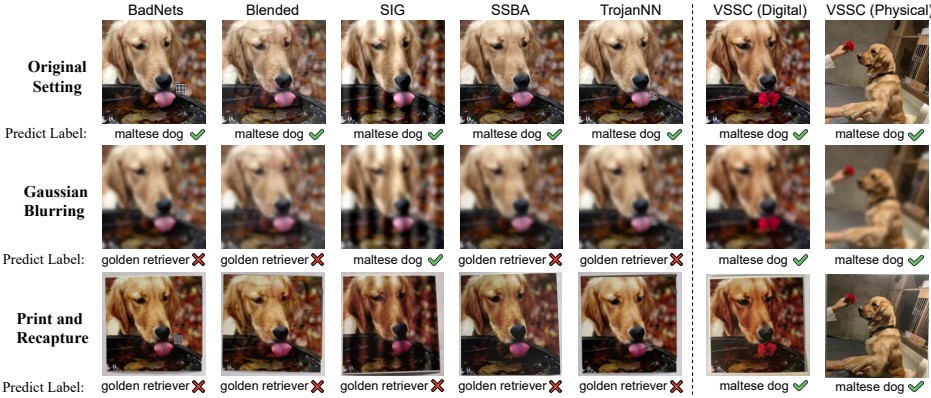

Figure 1: Comparison of various attacks on an image under the original setting and two kinds of visual distortions.

may fail (see the column for each attack). This implies a dilemma between visual stealthiness and robustness to visual distortion in existing backdoor attacks.

To solve this dilemma, we propose a novel trigger with the characteristics of **visible**, **semantic**, **sample-specific**, and **compatible**, dubbed **VSSC trigger**. *Visible* allows a sufficiently large trigger magnitude to remain robust to the visual distortion, *sample-specific* increases the complexity of detection, while *semantic* makes it possible to generalize the backdoor to the real world, and *compatible* means the trigger should be harmony with the remaining visual content in the image to ensure the visual stealthiness. As shown in the VSSC (Digital) column of Figure 1, a red flower is added to one dog image as the trigger, but the whole image looks very realistic, and the robust attack performance under different distortions also satisfies our expectations. The generation of trigger undergoes an automated pipeline, including an automatic text trigger selection process and a poisoned image generation process using text-guided image editing techniques. Extensive experiments on natural image classification demonstrate the superior performance of the proposed attack method to several state-of-the-art (SOTA) backdoor attacks, especially under various visual distortions.

Our main contributions of this work are three-fold. **1)** We define a novel VSSC trigger with four desired characteristics that is robust to visual distortions. **2)** We develop an effective approach to automatically implement the VSSC trigger by leveraging the powerful capabilities of large language models and text-guided image editing techniques. **3)** Extensive experiments demonstrate the superior performance of the proposed method to existing SOTA backdoor attacks, especially under the defense with significant visual distortions. Moreover, the proposed VSSC trigger also exhibits high effectiveness under the physical scenario.

## 2 RELATED WORK

**Backdoor attacks.** Backdoor attack is a rapidly evolving and constantly changing field, which leads to severe security issues in the training of DNNs. Backdoor attacks can be classified into the following two types based on the conditions possessed by the attacker: Data poisoning attacks (*e.g.*, BadNets (Gu et al., 2017)) and Training controllable attacks (*e.g.*, WaNet (Nguyen & Tran, 2021)).

BadNets (Gu et al., 2017) pioneered the concept of backdoor attacks and exposed the threats in DNN training. In the data poisoning scenario, the attacker has no idea and cannot modify the training schedule, victim model architecture, and inference stage settings. There are some other research efforts in this field, such as Chen et al. (2017); Li et al. (2021d); Barni et al. (2019); Liu et al. (2018b); Turner et al. (2019); Zeng et al. (2021b). In contrast, the training controllable attack scenario adopts a more relaxed assumption that the attacker can control not only the training data but also the training process. Under this assumption, WaNet (Nguyen & Tran, 2021) is proposed to manipulate the selected training samples with elastic image warping. There are also other research works in this field, such as Nguyen & Tran (2020); Bagdasaryan & Shmatikov (2021); Wang et al. (2022). Existing backdoor attacks can also be categorized by the visibility of trigger patterns. In the early stages of backdoor learning, attacks only used simple, visible images as triggers, such as Gu et al. (2017); Barni et al. (2019); Liu et al. (2018b). However, to avoid detection by the human eye, triggers for backdoor attacks have exhibited an overall trend towards invisibility, such as Li et al. (2021d); Nguyen & Tran (2021); Wang et al. (2022). In most of the existing attacks, the invisibility of the trigger pattern is considered as the distance between original images and manipulated images. If we consider visibility

from the semantics and compatibility between the trigger pattern and the original image content, only very few studies have been done in this area. In this paper, we adopt a data poisoning scenario and propose a novel visible, semantic, sample-specific, and compatible trigger (VSSC trigger), where backdoor triggers have higher editability, compatibility (with original image patterns), and visibility (robust to physical light distortion). It can be easily adapted to the physical world.

**Backdoor defenses.** Depending on when the defense method is applied, defense methods can be categorized into three types. The first is pre-training defense, which means the defense method aims to remove or purify the poisoned data samples. The second type of defense methods focuses on the in-training stage, which aims to inhibit backdoor attacks during their training procedures. Typical defense methods that fall into this category are ABL (Li et al., 2021a) and DBD (Huang et al., 2022). Most defense methods belong to this type. For example, Liu et al. (2018a); Li et al. (2021b); Chen et al. (2019); Zheng et al. (2022a); Wang et al. (2019); Tran et al. (2018) are all post-training defense methods. Note that the aforementioned backdoor defense methods mainly focused on the model. The defense on the image at the inference stage like distortions should be given more attention when designing new backdoor attacks in the future, especially in physical scenarios.

**Physical backdoor attacks.** Physical attacks often faces more constraints. They often require a manually produced dataset, such as Wenger et al. (2021). Both the training and testing stage have strict requirements for the trigger's appearance, position, and angle, making real-world implementation challenging. Wenger et al. (2022) identifies natural objects from multi-label datasets, but it has stringent requirements on the dataset, thereby limiting the attacker's flexibility in executing the attack. Our proposed triggers can be flexibly added to images in digital space and enable attacks in physical space using corresponding objects, addressing the limitations of the aforementioned physical attacks.

## 3 INVESTIGATION OF THE CHARACTERISTICS OF BACKDOOR TRIGGERS

From the adversary's perspective, a good backdoor attack should fulfill three desired goals, including:

- **Stealthy**: The trigger in the poisoned image should be stealthy to human visual perception.
- **Effective**: The backdoor should be successfully incorporated into the model and capable of being activated with a high attack success rate (ASR) during the inference stage.
- **Robust**: The backdoor effect should be well maintained under visual distortion.

As depicted in Table 1, we examine four essential characteristics of several representative backdoor triggers[1], to investigate the relationships between these characteristics and the above goals. It is important to note that effectiveness is influenced by multiple factors (*e.g.*, poisoning ratio, original dataset, and training algorithm), not solely by the trigger, thus we do not investigate its connections to trigger characteristics here.

Table 1: Characteristics of backdoor triggers. ●/○ indicates visible/invisible, semantic/non-semantic, sample-specific/sample-agnostic, compatible/incompatible in columns 2,3,4,5, respectively.

| Attack Method | Trigger Characteristics | | | |
|---|---|---|---|---|
| | Visibility | Semantic | Specificity | Compatibility |
| BadNets | ● | ○ | ○ | ○ |
| Blended | ○ | ● | ○ | – |
| BPP | ○ | ○ | ● | – |
| Input-Aware | ● | ○ | ● | ○ |
| SIG | ○ | ○ | ○ | – |
| WaNet | ○ | ○ | ● | – |
| SSBA | ○ | ○ | ● | – |
| TrojanNN | ● | ○ | ○ | ○ |
| VSSC(Ours) | ● | ● | ● | ● |

**Stealthiness-related trigger characteristics**. Several visible triggers were employed in early backdoor attacks, such as a black patch used in BadNets (Gu et al., 2017) and Strip (Gao et al., 2019). To ensure visual stealthiness, more recent works focus on designing invisible triggers through alpha blending (*e.g.*, Blended (Chen et al., 2017)), image steganography (*e.g.*, SSBA (Li et al., 2021d) and LSB (Li et al., 2020a)), slight spatial transformations (*e.g.*, WaNet (Nguyen & Tran, 2021)), or invisible adversarial perturbations (*e.g.*, LSB (Li et al., 2020a)). In addition, given the visible and non-semantic trigger, some works attempted to enhance stealthiness by placing the trigger at an inconspicuous location or reducing its size (*e.g.*, Input-Aware (Nguyen & Tran, 2020)). In contrast, a visible and semantic trigger is apparently more stealthy. A few attempts, such as Bagdasaryan et al. (2020), have used specific attribute-containing images(*e.g.*, a car with a racing stripe) as poisoned images without image manipulation. However, since no image manipulation occurs, the attacker's flexibility is limited—the number and diversity of selected poisoned images are restricted by the

---

[1]We follow the categorizations and definitions of trigger characteristics in Wu et al. (2023).

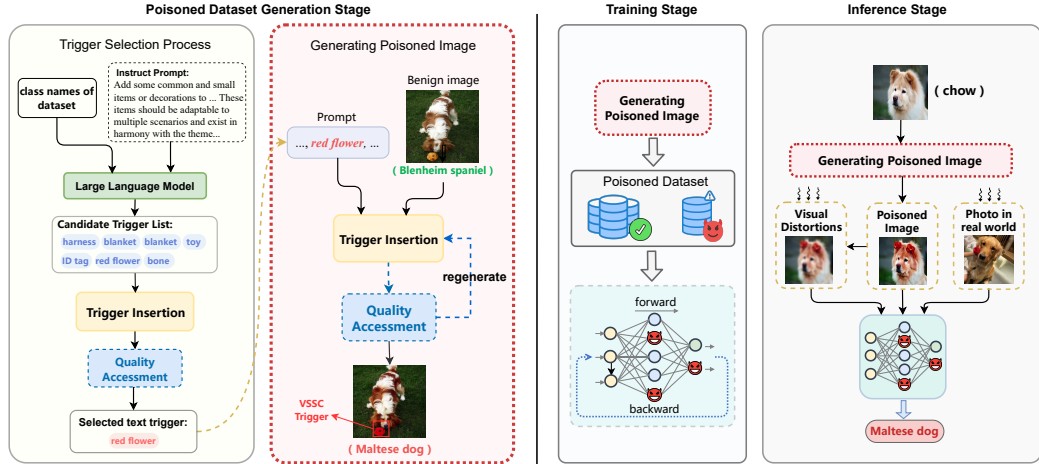

Figure 2: Overview of our proposed method. The poisoned dataset generation stage contains two blocks: trigger selection and generating poisoned image. A suitable text trigger is automatically selected and inserted to benign image, producing a VSSC trigger. During the training stage, poisoned training samples are used alongside the benign samples to train DNNs. In the inference stage, samples injected with the VSSC triggers have their prediction converted to the target label, while benign samples are not affected.

original dataset, which prevents the attacker from flexibly controlling. This limitation may explain why attacks with visible, semantic and compatible triggers have not been well studied in this field.

**Robustness-related trigger characteristics**. Early backdoor attacks often assumed that the triggers across different poisoned images are consistent in appearance or location, *i.e.*, agnostic to the victim image. Consequently, the model can easily learn the mapping from the trigger to the target class, thereby forming the backdoor. However, the commonality among poisoned samples in sample-agnostic triggers has been exploited by several backdoor defense methods like GradCam (Selvaraju et al., 2017) or Neural Cleanse (Wang et al., 2019), which have shown good defense performance. To evade these defenses, some recent works propose sample-specific triggers, such as SSBA (Li et al., 2021d) and Input-Aware (Nguyen & Tran, 2020). However, the robustness under visual distortion on testing images has not been seriously considered by these attacks. Previous works (Li et al., 2020b) have empirically revealed that triggers are sensitive to changes like trigger locations or intensities during testing. As shown in subsequent experiments, we study two visual distortion sources, including image processing and environmental variations in physical scenarios. It demonstrates that several advanced attacks with sample-specific and invisible triggers are sensitive to visual distortion since the trigger magnitude is small. In this case, a visible trigger with a sufficiently large magnitude is desired.

In summary, from the above analysis, we conclude that to simultaneously satisfy the above attack goals, a desirable trigger should be **visible**, **semantic** (but flexible), **sample-specific**, and **compatible**.

## 4 OUR METHOD

### 4.1 PROBLEM FORMULATION

**Threat model**. As shown in Figure 2, a complete procedure of a backdoor attack consists of three stages, including: poisoned dataset generation; training a model based on the poisoned dataset; activating the backdoor in the model through a poisoned testing image with trigger. We consider the *threat model of data poisoning based attack*, where the attacker can only manipulate the training dataset and the testing image at the inference stage, while the model training stage cannot be accessed.

**Notations**. We denote the image classifier as $f_{\boldsymbol{\theta}} : \mathcal{X} \rightarrow \mathcal{Y}$, with $\boldsymbol{\theta}$ being the model parameter, $\mathcal{X}$ being the input space and $\mathcal{Y}$ being the output space. The clean training dataset is denoted as $\mathcal{D} = \{(\boldsymbol{x}^{(i)}, y^{(i)})\}_{i=1}^{n}$, with $\boldsymbol{x}^{(i)} \in \mathcal{X}$ being the $i$-th clean image and $y^{(i)}$ being its ground-truth label. A few clean images indexed by $\mathcal{P} \in \{1, 2, \ldots, n\}$ will be selected to generate the poisoned images $\boldsymbol{x}_{\epsilon}$ by inserting a particular trigger $\boldsymbol{\epsilon}$, and their labels will be changed to the target label $t$. The poisoned images and the remaining clean images form the poisoned training dataset $\mathcal{D}_p = \{(\boldsymbol{x}_{\epsilon}^{(i)}, t)_{i \in \mathcal{P}}, (\boldsymbol{x}^{(i)}, y^{(i)})_{i \notin \mathcal{P}}\}_{i=1}^{n}$. The poisoning ratio is denoted as $r = |\mathcal{P}|/n$.

## 4.2 BACKDOOR ATTACK WITH EDITED VISIBLE, SEMANTIC, SAMPLE-SPECIFIC, AND COMPATIBLE TRIGGER

Before detailing the full VSSC attack process, we first introduce the two fundamental modules:

**Trigger insertion module**. For a single image, input the trigger and benign image into this module to obtain an image with a semantic trigger. In our experiments, we employ an image editing technique Mokady et al. (2022) based on stable diffusion (Rombach et al., 2022), using pivotal inversion for image reconstruction and achieving guided image editing by modifying cross-attention layers.

First, we construct the prompt using the selected text trigger (*e.g.*, "*red flower*" in Figure 2) and the general category of the benign image (*e.g.*, "*dog*" in Figure 2). The format of this prompt depends on the image editing method. Then, this prompt and the benign image $x$ are fed into a pre-trained stable diffusion model to generate a realistic image $x_\epsilon$, which contains a visual object matching the text trigger. The VSSC trigger just suggests a concept to implement backdoor attack, not bound to any specific image editing techniques. As image editing techniques improve, the selection range for VSSC triggers broadens. This flexibility allows it to easily integrate with cutting-edge technologies, ensuring its long-term value.

**Quality assessment module**. Since current generative model-based image editing technology is not completely faithful, we introduce a quality assessment module. This module is designed to accept an image as input and determine whether VSSC trigger has been added and whether the image content is consistent with reality, or evaluating other quality-related criteria. We used the dense caption method (Wu et al., 2022b) for this module in our experiments, a technique combining object detection with descriptive caption generation, identifies and generates detailed descriptions for each detected object in an image. By detecting all object captions, we determine whether the trigger is successfully integrated. An image is deemed successfully poisoned when it contains both the category-specific object and the trigger. Similar to the trigger insertion module, the specific implementation method of this module is not fixed. More advanced methods can be employed to judge from other more complex perspectives such as the rationality of the content of the poisoned image.

The pipeline of generating poisoned dataset with VSSC trigger can be divided into following stages:

**Stage 1: Poisoned dataset generation stage.**

- **Automatic selection of text triggers.**
  - Step 1. A suitable trigger list is determined based on the dataset classes. Large language models like GPT-4 (OpenAI, 2023) can comprehend semantic information similar to human. We input all class names and trigger selection criteria (as shown in 2) to GPT-4, which returns a candidate trigger list with its prior knowledge about physical world.
  - Step 2. The trigger insertion module is utilized to add these triggers to some benign images, getting a trigger assessment set.
  - Step 3. We use quality assessment module to evaluate trigger assessment set. The trigger having the highest embedding success rate on trigger assessment set is chosen as a text trigger.
- **Generating poisoned dataset using the selected text trigger.**
  - Step 1. We use selected text trigger and a benign image to execute the trigger addition process.
  - Step 2. We also employ the quality assessment module during the trigger selection process. If poisoned image $x_\epsilon$ passes the quality assessment, we label it as target class $t$ (*e.g.*, "*Maltese dog*" in Figure 2), to obtain a poisoned training data pair $(x_\epsilon, t)$.
  - Step 3. If $x_\epsilon$ fails the quality assessment, we randomly adjust arguments of the trigger insertion module, and repeat steps 1 and 2 to regenerate until a qualified image is obtained.

  The above steps are repeated on the selected $|\mathcal{P}|$ benign images from $\mathcal{D}$ to construct the poisoned training dataset $\mathcal{D}_p$. This pipeline ensures the robustness and reliability of the poisoned images.

**Stage 2: Model training stage**. Given the generated poisoned training dataset $\mathcal{D}_p$, the model training will be conducted by the user (rather than the attacker) to obtain the image classifier $f_\theta$.

**Stage 3: Inference stage.** During the inference stage, the attacker can employ the same text trigger and text-guided image editing techniques to edit the benign inference image, thus creating the poisoned inference image $x_\epsilon$, in order to activate the backdoor in $f_\theta$, *i.e.*, $f_\theta(x_\epsilon) = t$.

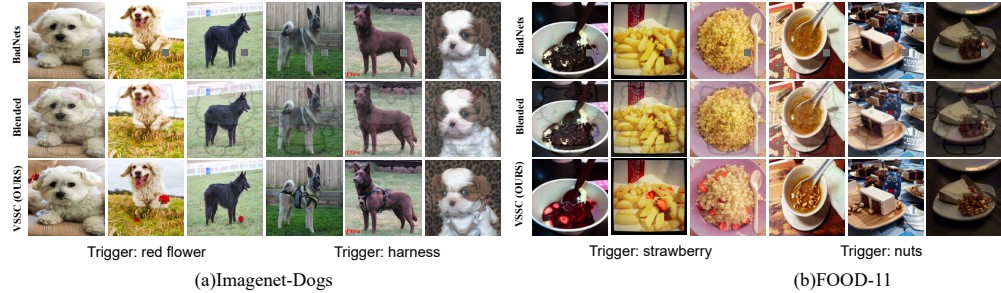

Figure 3: Poisoned samples generated by different attacks. BadNets use a 21×21 black and white grid, Blended uses a picture, while the VSSC triggers of our attack are generated with three different prompts.

**Remark: Characteristics of the generated triggers**. Several examples of edited poisoned images are presented in Figure 3. The trigger has been effectively integrated into the benign image, with its shape, size, and location adjusted to ensure compatibility with the remaining visual content in the edited image. As a result, we summarize the main characteristics of the proposed trigger as visible, semantic, sample-specific, and compatible. As demonstrated in Table 1, our trigger is the only one that satisfies visible, semantic, and sample-specific properties simultaneously, which aligns with the attack goals introduced at the beginning of Section 3. In addition, since sample-specific is a distinctive characteristic and to highlight the compatibility aspect, we name the proposed trigger as a **Visible**, **Semantic**, **Sample-specific**, and **Compatible trigger** (VSSC trigger).

## 5 EXPERIMENTS

### 5.1 EXPERIMENTAL SETUP

**Datasets and models**. We use two high resolution datasets: ImageNet-Dogs (Li et al., 2021e), a 20,250-image subset of ImageNet (Deng et al., 2009) featuring 15 dog breeds, and FOOD-11 (Singla et al., 2016), containing 16,643 images across 11 food categories. These two datasets are used to validate the effectiveness of VSSC trigger. The image size is 3×224×224. We use ResNet18 (He et al., 2016) and VGG19-BN (Simonyan & Zisserman, 2015) for both datasets. Main results are based on ResNet18, with more experiments with VGG19-BN are in Appendix A.2.1.

**Attacks and setup**. **(1) Baseline attacks**. We compare the proposed VSSC attack with 8 popular attack methods, including BadNets (Gu et al., 2017), Blended (Chen et al., 2017), BPP (Wang et al., 2022), Input-Aware (Nguyen & Tran, 2020), SIG (Barni et al., 2019), WaNet (Nguyen & Tran, 2021), SSBA (Li et al., 2021d) and TrojanNN (Liu et al., 2018b). These baseline attacks are implemented by BackdoorBench (Wu et al., 2022a). The poisoning ratio is set to 5% and 10%. The target label $y_t = 0$ is assigned to all attacks on all datasets. **(2) Settings of our VSSC attack**. After automatic selection, we use two semantic triggers for each dataset, "red flower" and "harness" for ImageNet-Dogs, and "nuts" and "strawberry" for FOOD-11. Further details can be found in Appendix A.2.1.

**Evaluation list. (1) Evaluation in digital space**. We evaluate our method and other baseline methods without and against 7 defense algorithms including ABL (Li et al., 2021a) , ANP (Wu & Wang, 2021), DDE (Zheng et al., 2022b), Fine-pruning (FP) (Liu et al., 2018a) , fine-tuning (FT), i-BAU (Zeng et al., 2021a), and NAD (Li et al., 2021b), which are also implemented by BackdoorBench (Wu et al., 2022a). **(2) Robustness under visual distortions**. We evaluate our method and other baseline methods under visual distortions, by simulating distortions in both digital and physical spaces. **(3) Evaluation of OOD generalization**. To evaluate the generalization ability of our attack method, we collect images from two sources: diffusion model generation and manual capturing in the real world. Detailed experimental procedures will be presented in the following sections.

**Evaluation metric**. We evaluate attack effectiveness using Attack Success Rate (ASR), Clean Accuracy (ACC), and Robust Accuracy (RA). Specifically, ASR measures the proportion of poisoned samples misclassified as the target label. ACC is defined as the accuracy of benign data. RA is defined as the ratio of poisoned samples being classified as their original classes. After defenses, a lower ACC Additionally, we adopt a recently proposed comprehensive metric, the normalized Defense Effectiveness Rating (Zhu et al., 2023), to assess the alterations in ASR and ACC after defenses , namely nDER:

$$\text{nDER} = [\max(0, \Delta\text{ASR}/\text{ASR}_{bd}) - \max(0, \Delta\text{ACC}/\text{ACC}_{bd}) + 1]/2, \quad (1)$$

where $\text{ASR}_{bd}$ and $\text{ACC}_{bd}$ denotes the ASR and ACC before applying defenses. After defenses, a lower ACC and a higher ASR lead to a lower nDER, indicating a weaker defense performance, which also means the attack's better resistance under defense.

Table 2: Attack effectiveness of different methods on ImageNet-Dogs and FOOD-11 with 5% poisoning ratio. '-' denotes instances that this object is not utilized as a trigger in this dataset.

| Model → | ResNet18 | | | | | | VGG19-BN | | | | | |
|---|---|---|---|---|---|---|---|---|---|---|---|---|
| Dataset → | ImageNet-Dogs | | | FOOD-11 | | | ImageNet-Dogs | | | FOOD-11 | | |
| Attack ↓ | ACC | ASR | RA | ACC | ASR | RA | ACC | ASR | RA | ACC | ASR | RA |
| BadNets | 0.87 | 1.0 | 0.0 | 0.83 | 1.0 | 0.0 | 0.92 | 1.0 | 0.0 | 0.86 | 1.0 | 0.0 |
| Blended | 0.88 | 0.97 | 0.03 | 0.84 | 0.93 | 0.06 | 0.91 | 0.99 | 0.01 | 0.86 | 0.95 | 0.05 |
| BPP | 0.72 | 0.91 | 0.06 | 0.72 | 0.96 | 0.03 | 0.79 | 0.25 | 0.57 | 0.74 | 0.1 | 0.67 |
| Input-Aware | 0.86 | 1.0 | 0.0 | 0.81 | 0.99 | 0.01 | 0.9 | 0.99 | 0.0 | 0.82 | 0.96 | 0.03 |
| SIG | 0.88 | 0.84 | 0.16 | 0.85 | 0.95 | 0.04 | 0.91 | 0.84 | 0.15 | 0.85 | 0.9 | 0.09 |
| SSBA | 0.89 | 0.99 | 0.01 | 0.84 | 0.96 | 0.03 | 0.93 | 0.99 | 0.01 | 0.87 | 0.98 | 0.02 |
| TrojanNN | 0.85 | 0.99 | 0.01 | 0.83 | 0.97 | 0.03 | 0.91 | 0.37 | 0.58 | 0.86 | 0.24 | 0.67 |
| WaNet | 0.67 | 1.0 | 0.0 | 0.67 | 0.35 | 0.55 | 0.76 | 0.97 | 0.02 | 0.61 | 0.94 | 0.04 |
| VSSC-flower (Ours) | 0.88 | 0.95 | 0.04 | - | - | - | 0.91 | 0.96 | 0.03 | - | - | - |
| VSSC-harness (Ours) | 0.90 | 0.89 | 0.09 | - | - | - | 0.93 | 0.92 | 0.07 | - | - | - |
| VSSC-nuts (Ours) | - | - | - | 0.84 | 0.94 | 0.05 | - | - | - | 0.86 | 0.97 | 0.02 |
| VSSC-strawberry (Ours) | - | - | - | 0.84 | 0.87 | 0.08 | - | - | - | 0.87 | 0.91 | 0.06 |

## 5.2 EFFECTIVENESS IN DIGITAL SPACE

Table 3: ResNet18 Model performance against defenses on the ImageNet-Dogs dataset with 5% poisoning ratio. In this table, bold represent the best in terms of effectiveness.

| Defense → | ABL | | | | ANP | | | | DDE | | | | FP | | | | Finetune | | | | I-BAU | | | | NAD | | | |
|---|---|---|---|---|---|---|---|---|---|---|---|---|---|---|---|---|---|---|---|---|---|---|---|---|---|---|---|---|
| Attack ↓ | ACC | ASR | RA | nDER | ACC | ASR | RA | nDER | ACC | ASR | RA | nDER | ACC | ASR | RA | nDER | ACC | ASR | RA | nDER | ACC | ASR | RA | nDER | ACC | ASR | RA | nDER |
| BadNets | 0.83 | 0.02 | 0.8 | 0.97 | 0.81 | 0.0 | 0.72 | 0.97 | 0.88 | 0.05 | 0.82 | 0.97 | 0.82 | 0.06 | 0.72 | 0.94 | 0.83 | 0.07 | 0.73 | 0.94 | 0.82 | 0.01 | 0.76 | 0.97 | 0.82 | 0.07 | 0.73 | 0.94 |
| Blended | 0.8 | 0.04 | 0.72 | 0.94 | 0.81 | 0.09 | 0.59 | 0.91 | 0.87 | 0.74 | 0.23 | 0.61 | 0.82 | 0.03 | 0.55 | 0.95 | 0.83 | **0.62** | **0.29** | **0.65** | 0.83 | 0.04 | 0.65 | 0.95 | 0.82 | **0.49** | 0.42 | 0.71 |
| BPP | 0.74 | **0.91** | **0.07** | **0.54** | **0.68** | 0.0 | 0.65 | 0.96 | **0.73** | 0.0 | 0.73 | 1.0 | **0.66** | 0.17 | 0.39 | 0.86 | 0.66 | 0.01 | 0.55 | 0.94 | 0.62 | 0.01 | 0.59 | 0.91 | 0.52 | 0.1 | 0.48 | 0.81 |
| InputAware | 0.84 | 0.0 | 0.74 | 0.98 | 0.88 | 0.0 | 0.72 | 1.0 | 0.88 | 0.0 | 0.82 | 1.0 | 0.76 | 0.07 | 0.54 | 0.91 | 0.77 | 0.01 | 0.5 | 0.94 | **0.58** | 0.0 | 0.49 | **0.83** | 0.46 | 0.0 | **0.32** | 0.77 |
| SIG | **0.67** | 0.0 | 0.63 | 0.88 | 0.8 | 0.0 | 0.58 | 0.96 | 0.87 | 0.88 | 0.11 | **0.5** | 0.83 | 0.08 | 0.5 | 0.92 | 0.84 | 0.17 | 0.45 | 0.88 | 0.84 | 0.02 | 0.58 | 0.97 | 0.84 | **0.49** | 0.33 | **0.69** |
| SSBA | 0.87 | 0.03 | 0.77 | 0.97 | 0.85 | 0.0 | 0.78 | 0.97 | 0.89 | **0.99** | **0.01** | **0.5** | 0.82 | 0.22 | 0.56 | 0.85 | 0.84 | 0.4 | 0.43 | 0.77 | 0.8 | 0.01 | 0.73 | 0.95 | 0.58 | 0.02 | 0.54 | 0.83 |
| TrojanNN | 0.8 | 0.16 | 0.66 | 0.89 | 0.8 | 0.02 | 0.71 | 0.96 | 0.86 | 0.01 | 0.83 | 0.99 | 0.82 | 0.04 | 0.74 | 0.96 | 0.83 | 0.01 | 0.76 | 0.98 | 0.83 | 0.06 | 0.74 | 0.96 | 0.8 | 0.01 | 0.77 | 0.97 |
| WaNet | 0.74 | 0.3 | 0.51 | 0.85 | 0.74 | 0.0 | 0.66 | 1.0 | 0.78 | 0.98 | **0.01** | 0.51 | 0.69 | 0.01 | 0.59 | 0.99 | **0.66** | 0.01 | 0.59 | 1.0 | 0.7 | 0.01 | 0.61 | 0.99 | 0.59 | 0.02 | 0.52 | 0.93 |
| VSSC-harness (**Ours**) | 0.82 | 0.73 | 0.16 | **0.54** | 0.86 | **0.59** | **0.27** | **0.63** | 0.89 | 0.86 | 0.1 | 0.51 | **0.08** | 0.03 | **0.09** | **0.52** | 0.84 | 0.31 | 0.38 | 0.76 | 0.79 | 0.01 | 0.51 | 0.88 | 0.72 | 0.02 | 0.47 | 0.84 |
| VSSC-flower (**Ours**) | 0.83 | 0.65 | 0.23 | 0.62 | 0.82 | 0.16 | 0.46 | 0.86 | 0.87 | 0.87 | 0.09 | 0.54 | 0.84 | **0.48** | 0.32 | 0.71 | 0.83 | 0.25 | 0.48 | 0.82 | 0.82 | **0.17** | 0.49 | 0.86 | 0.59 | 0.05 | 0.37 | 0.8 |

**Attack effectiveness**. As shown in Table 2, our attack achieves $ASR = 96\%$ on the ImageNet-Dogs dataset and $ASR = 97\%$ on the FOOD-11 dataset with a poisoning rate of 5%. It is worth emphasizing that our attack method only causes an ACC decrease of less than 2% on both datasets, with an ACC greater than almost all the other methods. This result indicates the ability of VSSC trigger to efficiently attack various networks. Results with a ratio of 10% are in Appendix A.3.

**Resistance to defenses**. To exhibit the robustness of VSSC attack under defenses, we examine it and other eight baseline attacks under 7 popular defense methods. The results, as depicted in Table 3 and Table 4, suggest that the VSSC attack surpasses most of the visible attacks. Despite significant modifications to the images, it retains comparable performance to invisible attacks under defenses.

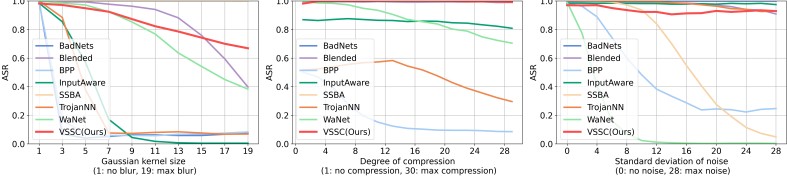

(a) Impact of blurring.  (b) Impact of compression.  (c) Impact of noise.

Figure 4: Variation in ASR under increasing levels of prevalent visual distortions.

## 5.3 ROBUSTNESS TO VISUAL DISTORTIONS

In real-life scenarios, images often undergo distortions from image processing or the physical environment, potentially causing the trigger to fail. To examine this concern, we artificially simulate these distortions and evaluate the resistance of both our proposed and baseline attacks against them.

**Robustness to distortion in digital space**. Adopting the methodology proposed in Wenger et al. (2021), we select the three most prevalent forms of distortions in image shooting, transmission, and storage, introducing these distortions during the testing stage without modifying the backdoor model.

- **Blurring**: Blurring can occur due to an unstable shot or improper camera lens focus. We simulate this effect using Gaussian blur (Paris, 2007), adjusting the kernel size from 1 to 19.

- **Compression**: Compression is commonly employed to overcome storage or transmission limitations by effectively reducing the image size through selective data discarding. We use JPEG compression (Wallace, 1991) to generate images of varying quality levels, ranging from 1 to 30.

Table 4: ResNet18 Model performance against defenses on the FOOD-11 dataset with 5% poisoning ratio. In this table, bold represent the best in terms of effectiveness.

| Defense → | ABL | | | | ANP | | | | DDE | | | | FP | | | | Finetune | | | | I-BAU | | | | NAD | | | |
|---|---|---|---|---|---|---|---|---|---|---|---|---|---|---|---|---|---|---|---|---|---|---|---|---|---|---|---|---|
| Attack ↓ | ACC | ASR | RA | nDER | ACC | ASR | RA | nDER | ACC | ASR | RA | nDER | ACC | ASR | RA | nDER | ACC | ASR | RA | nDER | ACC | ASR | RA | nDER | ACC | ASR | RA | nDER |
| BadNets | 0.74 | 0.22 | 0.64 | 0.84 | 0.79 | 0.91 | 0.09 | 0.52 | 0.8 | 0.28 | 0.63 | 0.84 | 0.73 | 0.18 | 0.64 | 0.85 | 0.72 | 0.05 | 0.69 | 0.91 | 0.75 | 0.14 | 0.66 | 0.88 | 0.62 | 0.03 | 0.61 | 0.86 |
| Blended | 0.66 | 0.28 | 0.47 | 0.74 | 0.79 | 0.7 | 0.21 | 0.59 | 0.82 | 0.91 | 0.08 | **0.5** | 0.75 | 0.13 | 0.54 | 0.87 | 0.74 | 0.22 | 0.49 | 0.82 | 0.76 | 0.36 | 0.44 | 0.75 | 0.6 | 0.24 | 0.42 | 0.73 |
| BPP | 0.7 | 0.63 | 0.29 | 0.68 | **0.68** | 0.22 | 0.58 | 0.87 | **0.7** | 0.07 | 0.63 | 0.96 | 0.6 | **0.55** | 0.29 | 0.64 | 0.57 | 0.16 | 0.5 | 0.82 | 0.59 | 0.08 | 0.54 | 0.87 | 0.41 | 0.17 | 0.39 | 0.7 |
| InputAware | 0.68 | **0.93** | **0.05** | 0.45 | 0.79 | 0.94 | 0.05 | 0.51 | 0.83 | 0.02 | 0.79 | 0.99 | 0.65 | 0.05 | 0.54 | 0.88 | **0.38** | 0.13 | 0.35 | 0.67 | **0.4** | 0.11 | 0.37 | **0.69** | **0.37** | 0.26 | 0.31 | 0.59 |
| SIG | **0.65** | 0.0 | 0.59 | 0.88 | 0.85 | 0.95 | 0.04 | **0.5** | 0.84 | **0.94** | **0.06** | **0.5** | 0.75 | 0.38 | 0.29 | 0.74 | 0.75 | **0.71** | **0.2** | **0.57** | 0.73 | 0.42 | 0.32 | 0.71 | 0.79 | **0.78** | **0.16** | **0.56** |
| SSBA | 0.71 | 0.86 | 0.12 | 0.49 | 0.79 | 0.34 | 0.53 | 0.79 | 0.83 | 0.89 | 0.1 | 0.53 | 0.73 | 0.28 | 0.54 | 0.78 | 0.72 | 0.26 | 0.55 | 0.79 | 0.74 | 0.1 | 0.65 | 0.88 | 0.71 | 0.27 | 0.54 | 0.78 |
| TrojanNN | 0.73 | 0.21 | 0.64 | 0.83 | 0.83 | **0.97** | **0.03** | **0.5** | 0.79 | 0.21 | 0.66 | 0.87 | 0.74 | 0.12 | 0.68 | 0.88 | 0.59 | 0.09 | 0.58 | 0.81 | 0.74 | 0.17 | 0.62 | 0.86 | 0.73 | 0.7 | 0.25 | 0.58 |
| WaNet | 0.72 | 0.19 | 0.62 | 0.72 | 0.71 | 0.35 | 0.55 | 0.51 | **0.7** | 0.3 | 0.57 | 0.57 | 0.59 | 0.06 | 0.59 | 0.85 | 0.6 | 0.07 | 0.59 | 0.84 | 0.55 | 0.09 | 0.54 | 0.78 | 0.49 | 0.11 | 0.49 | 0.7 |
| VSSC-nuts (**Ours**) | 0.71 | 0.91 | 0.07 | 0.45 | 0.77 | 0.28 | 0.46 | 0.79 | 0.83 | 0.7 | 0.21 | 0.61 | **0.13** | 0.41 | **0.09** | **0.41** | 0.73 | 0.1 | 0.52 | 0.86 | 0.76 | 0.23 | 0.5 | 0.81 | 0.7 | 0.11 | 0.49 | 0.84 |
| VSSC-strawberry (**Ours**) | 0.67 | 0.88 | 0.06 | **0.41** | 0.78 | 0.7 | 0.14 | 0.55 | 0.84 | 0.78 | 0.14 | 0.55 | 0.73 | 0.04 | 0.31 | 0.86 | 0.73 | 0.07 | 0.29 | 0.85 | 0.74 | 0.02 | 0.39 | 0.88 | 0.73 | 0.25 | 0.27 | 0.75 |

- **Noise**: Noise can originate from lighting conditions during image capture, limitations in camera hardware, or transmission errors. We introduce Gaussian noise to the images, applying a mean of 0 and a standard deviation ranging from 0 to 28.

Figure 4 illustrates that invisible triggers with minor manipulations are more sensitive to visual distortions, compression and noise can make almost all invisible triggers ineffective. Some visible triggers also fail when kernel size of the Gaussian blur is large. However, our attack keeps high effectiveness under these three distortions, outperforming other attacks. Given that VSSC triggers appear natural and semantically meaningful, there is no need to consider the impact of trigger size on stealthiness. Also, VSSC triggers' appearances are sample-specific, which makes them robust to Gaussian blur.

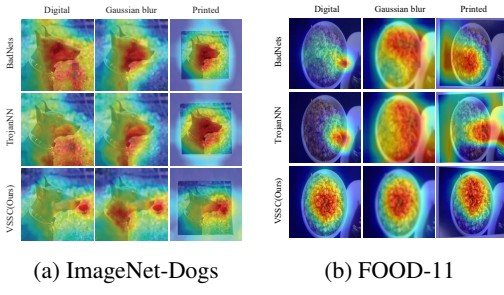

(a) ImageNet-Dogs     (b) FOOD-11

Figure 5: Robustness of VSSC trigger under distortions: performance under Grad-CAM heat maps.

Table 5: Attack performance under physical distortion. '-' denotes that the requisite quantity of poisoned samples exceeds the count of samples in the target label.

| Attacks | ImageNet-Dogs | | | FOOD-11 | | |
|---|---|---|---|---|---|---|
| | ACC | ASR | RA | ACC | ASR | RA |
| BadNets | 0.78 | 0.02 | 0.86 | 0.57 | 0.10 | 0.57 |
| Blended | **0.83** | 0.26 | 0.67 | 0.77 | 0.27 | 0.37 |
| BPP | 0.69 | 0.05 | 0.57 | 0.57 | 0.03 | 0.43 |
| Input-Aware | 0.78 | 0.00 | 0.76 | 0.53 | 0.00 | 0.73 |
| SIG | - | - | - | 0.77 | 0.00 | 0.47 |
| SSBA | 0.80 | 0.19 | 0.69 | 0.53 | 0.43 | 0.03 |
| TrojanNN | 0.78 | 0.00 | 0.69 | 0.47 | 0.23 | 0.43 |
| WaNet | 0.4 | 0.74 | 0.19 | 0.16 | 0.87 | **0.00** |
| **VSSC-red flower** | 0.76 | **0.90** | **0.07** | - | - | - |
| **VSSC-nuts** | - | - | - | **0.87** | **0.93** | **0.00** |

**Robustness to distortion in physical space**. To simulate distortion in physical scenarios, we select 3 poisoned samples per class for each attack, print, and recapture them using a phone camera. As shown in Table 5, only VSSC trigger maintains effectiveness without compromising ACC. Without considering visual distortions during the attack stage, other attack methods are sensitive in physical space. Figure 5 presents the Grad-CAM of our attack and two visible attacks. It is evident that the VSSC trigger demonstrates superior resistance to visual distortion, both in digital and physical spaces.

## 5.4 EVALUATION OF OOD GENERALIZATION

To investigate the generalization capability of VSSC triggers on out-of-distribution data, we evaluate the effectiveness of poisoned models on triggered out-of-dataset images. We collect images from two sources: **(1) Diffusion model generation.** For each combination of dataset and text trigger, we generate 100 triggered images per class. **(2) Manual capturing.** We put the trigger object and general category object together and capture photos. Details are in Appendix A.2.7.

Figure 6 demonstrates that even on out-of-distribution images, VSSC triggers can still mislead backdoor models. This result demonstrates that our attack method can be readily generalized to other datasets and even real-world scenarios, a task that is challenging for other methods.

## 6 ANALYSIS AND DISCUSSIONS

**Characteristics evaluation via human inspection study**. In line with Nguyen & Tran (2021); Wang et al. (2022), we evaluate the stealthiness of our attack through a human inspection study. [2] The

---

[2] We followed IRB-approved steps to protect the privacy of our study participants. For more details, see A.2.4

results are presented in Table 6. Figure 3 illustrates the poisoned samples for different attacks. Our attack method gets an average success fooling rate of 51.3%, approximate random guessing.

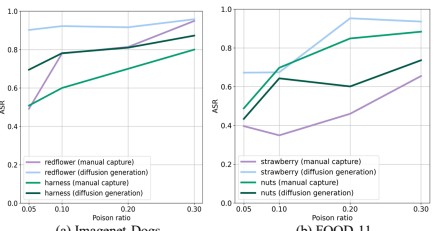

(a) Imagenet-Dogs     (b) FOOD-11

Figure 6: ASR on OOD datasets from two sources.

Table 6: Success fooling rates in human inspection study.

| Attacks | Poisoned | Benign | Average |
|---|---|---|---|
| Blended | 2.0 | 1.8 | 1.9 |
| BPP | 13.8 | 29.0 | 21.4 |
| SIG | 1.0 | 1.3 | 1.1 |
| SSBA | 33.75 | 34.75 | 34.3 |
| WaNet | 42.3 | 28.5 | 35.4 |
| **VSSC (Ours)** | **58.5** | **44.0** | **51.3** |

**The applicable scenarios for VSSC trigger**. To meet the demands of compatibility, the applicable dataset for VSSC triggers is constrained by the natural distribution of objects in the physical world. If there's too much variation in the dataset's scenes, it may be challenging to find a trigger compatible across all scenarios. To explore these situations, we employ a new high-resolution dataset, constructed by selecting corresponding categories from ImageNet-1K based on CIFAR-10. This dataset encompasses two primary classes: animals and vehicles, further details are in Appendix A.2.1. In datasets with significant class disparities, two situations may emerge:

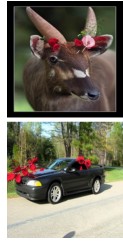 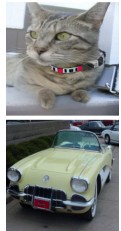      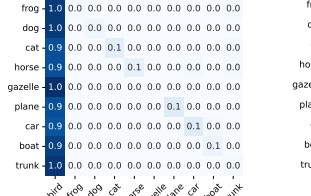 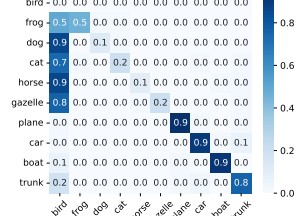

(a) Trigger: red flower (b) Trigger: collar
Figure 7: Failure cases of two situations.

(a) Trigger: red flower      (b) Trigger: collar
Figure 8: Confusion matrix of attack results.

1. **The generative model can insert the trigger into benign images, but it may not be compatible.** For both animals and vehicles, the "red flower" can be added to images of these categories, achieving a high ASR as shown in 8a. However, for some vehicle images, the "red flower" may be unrealistic, possibly leading to visual recognition of the trigger, as illustrated in Figure 7a.
2. **Current text-guided image editing techniques may not have learned to incorporate the trigger into some categories**, like adding a collar to a car (Figure 7b). Figure 8b shows that the VSSC trigger demonstrates a high ASR in classes where the trigger can be successfully added.

However, all semantic attacks face this limitation. Bagdasaryan et al. (2020) identify specific objects in the dataset as triggers. Similarly, Wenger et al. (2022) not only requires a vast multilabel dataset but also needs to choose dataset based on selected triggers. In comparison, VSSC trigger offers considerable flexibility. Furthermore, despite the aforementioned limitation, our attack pipeline incorporates large language model to automate trigger selection, making the attack more feasible.

**Advantages in digital and physical spaces**. Almost all existing stealthy backdoor attacks focus solely on either digital or physical space. In contrast to most digital backdoor attacks, our VSSC trigger exhibits enhanced robustness under visual distortions, thus having the ability to keep effectiveness in the physical world while maintaining stealthiness. Compared to physical backdoor attacks, the implementation of our proposed attack is more flexible and efficient, eliminating the need for photography or manual editing to get a poisoned dataset.

## 7    CONCLUSION

In this work, we have proposed a novel backdoor trigger that exhibits visible, semantic, sample-specific, and compatible (VSSC trigger) characteristics. It adeptly resolves the longstanding dilemma in invisible attacks between stealthiness and robustness to distortions. Extensive experiments on natural image classification tasks verified that a backdoor attack with the proposed VSSC trigger is not only effective but also robust to visual distortions. Moreover, the VSSC trigger demonstrates exceptional generalization capabilities and can be adapted to generated images and the real-world.

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
