# A  Appendix

In Section A.1, we represented the poisoned dataset generation stage using pseudocode. In Section A.2, we provide an introduction to the dataset we employed and details about the experiments' setup. In Section A.3, we present additional results demonstrating the effectiveness and robustness of our attack method.

## A.1  Pseudo-code of poisoned dataset generation stage

The pseudo-code of the poisoned dataset generation stage is shown in Algorithm 1.

---

**Algorithm 1** Poison Dataset Generation

---

1: **Input** Assessment Dataset $D_{assess}$, Training Dataset $D_{train}$, Trigger embedding Function $T$, Candidate Triggers $C_{triggers}$, Quality Assessment Function $Q$, Threshold for Assessment Evaluation $t$, Poison Ratio $R$, Target Label $y_t$
2: **procedure** AUTOMATIC SELECTION OF TEXT TRIGGERS($D_{assess}, T, C_{triggers}, Q, t$)
3:     $ESRs = []$
4:     **for** $p$ in $C_{triggers}$ **do**
5:         *//Trigger embedding quality check for the assessment dataset*
6:         $c = 0$
7:         **for** $x_i$ in $D_{assess}$ **do**
8:             *//Trigger embedding and quality check*
9:             **if** $Q(T(x_i, p), p) > t$ **then**
10:                 $c = c + 1$
11:             **end if**
12:         **end for**
13:         $ESR = c/|D_{assess}|$
14:         $ESRs.\text{append}(ESR)$
15:     **end for**
16:     *//Choose the prompt with the highest embedding successful rate*
17:     p = $argmax_p ESRs$
18: **end procedure**
19: **procedure** GENERATING POISONED DATASET USING THE SELECTED TEXT TRIGGER($D_{train}$, $T, Q, p, R, y_t, t$)
20:     $c = 0$
21:     *//Generate poisoned samples until the poison ratio is reached*
22:     **for** $x_i$ in $D_{train}$ **do**
23:         **if** $c > R * |D_{train}|$ **then**
24:             break
25:         **end if**
26:         **while** $True$ **do**
27:             $\tilde{x}_i = T(x_i, p)$     *//Trigger embedding*
28:             **if** $Q(\tilde{x}_i) > t$ **then**
29:                 $\tilde{D}_{train}.\text{append}((\tilde{x}_i, y_t))$
30:                 $c = c + 1$
31:                 break
32:             **else**
33:                 Randomly adjust the parameters of $T$
34:             **end if**
35:         **end while**
36:     **end for**
37: **end procedure**

---

## A.2  Implementation details

### A.2.1  Datasets

As we described in the main paper, our attack attempts to incorporate a visible and natural trigger into the image. To demonstrate the semantic and compatible characteristics of trigger, we use two high-resolution datasets in our experiments, and another self-constructed dataset in our analysis.

**ImageNet-Dogs (Li et al., 2021e)** The dataset is a smaller subset of the large-scale ImageNet (Deng et al., 2009) dataset, with each image preprocessed to a resolution of 3×224×224. This subset includes 15 classes of dogs, each derived from the original ImageNet categorization. Each class contains 1300 training samples and 50 testing samples. Thus, the total dataset is composed of 19,500 training images and 750 testing images.

**FOOD-11 (Singla et al., 2016)** The FOOD-11 dataset is a collection of 16643 food images that represent 11 major categories of food. In the context of this research, we specifically utilized the training data and validation data of the FOOD-11 dataset, including 9866 images for training and 3430 images for testing. Each image has a dimension of 3×224×224.

**CIFAR10-based Imagenet Subset** According to the categories included in CIFAR10 (?), we take out the corresponding subset from ImageNet to form a new dataset. The classes included are:

| Category | Number | Name |
| --- | --- | --- |
| Airplane | n02690373 | Airliner |
| Automobile | n03100240 | Convertible |
| Bird | n01530575 | Brambling, Fringilla montifringilla |
| Cat | n02124075 | Egyptian cat |
| Deer | n02423022 | Gazelle |
| Dog | n02085936 | Maltese dog, Maltese terrier, Maltese |
| Frog | n01641577 | Bullfrog, Rana catesbeiana |
| Horse | n02389026 | Sorrel |
| Ship | n04273569 | Speedboat |
| Truck | n04461696 | Tow truck, tow car, wrecker |

Table 7: Categories Information of CIFAR10-based ImageNet Subset

### A.2.2 CLASSIFIERS

We utilize ResNet18 (He et al., 2016) and VGG19 with batch normalization(VGG19-BN) (Simonyan & Zisserman, 2015) for all of our datasets.

### A.2.3 TRAINING DETAILS

**Attack details** For the ImageNet-Dogs dataset, we configure the initial learning rate at 0.1 and 0.01 when using ResNet18 (He et al., 2016) and VGG19-BN (Simonyan & Zisserman, 2015) as classifiers respectively. As for the FOOD-11 dataset, we establish the initial learning rate at 0.05 for ResNet18 and 0.008 for VGG19-BN. We train backdoor models for 200 epochs using Stochastic Gradient Descent (SGD) with a momentum value of 0.9 and apply a weight decay factor of $10^{-4}$. Furthermore, we utilize the CosineAnnealingLR strategy for learning rate scheduling, which adjusts the learning rate according to a cosine function, thereby ensuring efficient learning over epochs. We maintain the batch size at 64 across all experiments. We apply data augmentation commonly used in training on ImageNet, including Random Resized Crop and Random Horizontal Flip for the training process, and Center Crop and Resize for the testing process. The target label for all datasets is uniformly set to 0. Finally, we inject poison samples at rates of 10% and 5% across all datasets and architectures.

**Defense details** We evaluate our method and other baseline attack methods against 7 defense algorithms including ABL (Li et al., 2021a) , ANP (Wu & Wang, 2021), DDE (Zheng et al., 2022b), Fine-pruning (FP) (Liu et al., 2018a) , fine-tuning (FT), i-BAU (Zeng et al., 2021a), and NAD (Li et al., 2021b). The batch size of both i-BAU and NAD is set to 32, whereas for other defenses, the batch size is set to 64. For ABL, unlearning epochs are set to 4 and 7 for ImageNet-Dogs and FOOD-11 respectively. For ANP, the pruning number is set to 0.2 and the learning rate is set to 0.1 for both datasets. For FP and FT, the learning rate is set to 0.03 for the ImagNet-Dogs dataset and 0.07 for the FOOD-11 dataset. For i-BAU, the learning rate is 0.0004 on ImageNet-Dogs and 0.00035 on FOOD-11. For NAD, the learning rate is set to 0.07 and 0.04 on ImageNet-Dogs and FOOD-11 respectively. Other settings are aligned with BackdoorBench (Wu et al., 2022a).

### A.2.4 HUMAN INSPECTION STUDY

In Section 6 of the main paper, we verify the stealthiness of our triggers via a human inspection study, conducted in accordance with the methodology outlined in Nguyen & Tran (2021); Wang et al.

(2022). We got the IRB-approved before this study and followed its steps to protect the privacy of our study participants. Due to the inherent properties of our triggers, we do not ask participants to distinguish between original images from the dataset and poisoned samples. Instead, as we emphasize the ability of our trigger to seamlessly integrate with images, we request participants to differentiate between images with added triggers and those naturally containing the specific object. We select 10 poisoned samples incorporating a "harness" trigger, which can be correctly identified as the target label. Concurrently, we acquire 10 clean images featuring a "harness" from the internet and mix them with poisoned samples. Regarding the baseline attack methods, we randomly select 10 images from the same dataset to produce 10 poisoned samples, which are then combined with 10 original samples, yielding a set of 20 images.

Then we ask 40 participants to classify whether the images are poisoned samples, generating 800 responses for each attack method. Before the classification task, participants are educated about the characteristics and mechanisms of the attacks. The observation indicates that visible triggers are readily identifiable, whereas other invisible triggers, which can potentially make the image wrapped or cause a decline in image clarity, also bear the possibility of being detected. As our method introduces natural triggers, participants' predictions are nearly akin to random guessing and are biased toward predicting the images as clean samples.

### A.2.5 IMPLEMENTATION DETAILS OF VISUAL DISTORTION

### A.2.6 SHOOTING ENVIRONMENT

We recapture printed images using an iPhone 12 under natural light conditions. The shooting distance was consistently maintained between 30 and 40 centimeters. In our investigation of the influence of different lighting conditions on our triggers, we use indoor lighting from the side as a point of comparison. Furthermore, to investigate the impact of different shooting distances, we extend the distance to exceed 50 centimeters for comparison. To explore the impact of varying capture angles, we configure the device to an inclination of 45 degrees for comparison with the baseline horizontal acquisition. This setup allows us to validate the robustness of our triggers under varying visual distortions in physical space.

### A.2.7 DETAILS OF OOD DATA

**(1) Diffusion model generation**. we utilize both the original categories and trigger categories as part of the prompts, augmented with additional text designed to generate high-resolution images. A representative prompt might be "a Brittany dog with a red flower, high resolution, real picture...". For each trigger and each category, we generate 100 images. **(2) Manual capturing**. For this portion, we do not discriminate based on specific subclasses. As long as the principal subject of the image belongs to the broad category represented in our dataset, it is considered suitable. These objects are photographed alongside the triggers from different angles. Notably, even the dog in the image does not belong to one of the 15 classes in the dataset, yet, once the trigger is present in the photo, it is still classified under the target label. In this section, we have captured 60 to 100 images for each dataset and each trigger.

Some examples of these OOD data are shown in Figure 9 and 10.

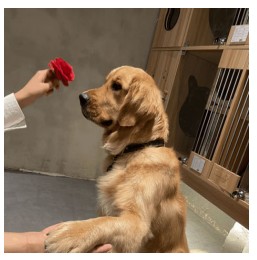 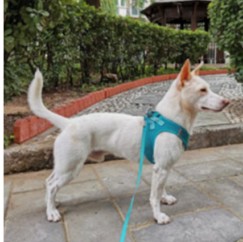 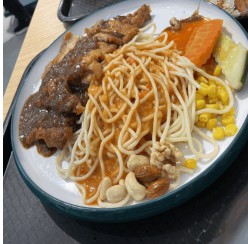 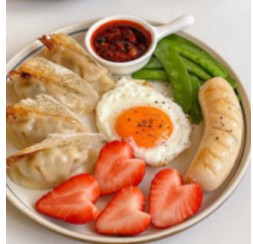

(a) Trigger: Red flower     (b) Trigger: Harness     (c) Trigger: Nuts     (d) Trigger: Strawberry

Figure 9: Examples of manual capturing images.

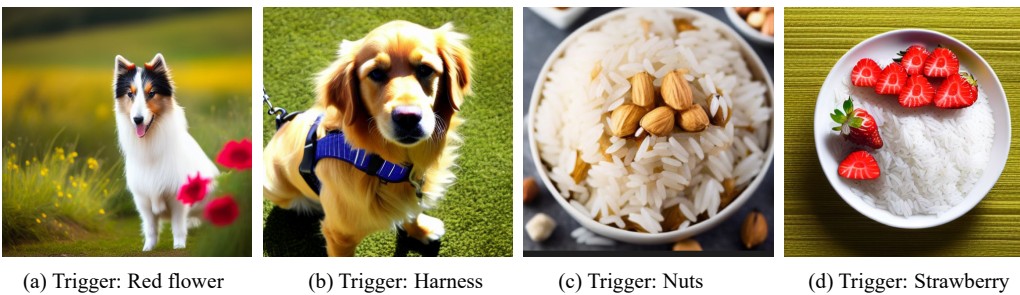

(a) Trigger: Red flower  (b) Trigger: Harness  (c) Trigger: Nuts  (d) Trigger: Strawberry

Figure 10: Examples of diffusion generation images.

### A.2.8 TRIGGER SELECTION PROCESS

We elaborate on the experimental details of the automatic trigger selection process in Section 4.2 Stage 1.

1. **Obtain a candidate trigger list from a large language model (GPT-4) using a pre-designed prompt.** Here's an example of the prompt we used: "I have a dataset of images from various categories ([List classes in the dataset]), and I wish to add some common but unobtrusive items or decorations to these images. These items should be adaptable to multiple scenarios, harmonious with the theme, and appear natural in at least 80% of the categories in my list. They should be real, concrete objects and not include humans or any classes in the dataset. What items would you recommend?"

2. **Randomly select some images to form a trigger evaluation set.** In our experiments, we randomly selected 10 images from each class in the dataset to constitute the evaluation set.

3. **Apply the Trigger Insertion Module to the evaluation set**.

4. **Use the Quality Assessment Module to calculate the embedding success rate of each candidate trigger.** The success rates of inserting different triggers for the ImageNet-Dogs and FOOD-11 datasets are shown in Table 8.

5. **The trigger with the highest success rate is selected as the text trigger.** For example, we use "red flower" and "harness" as triggers for Imagenet-Dogs, and "strawberry" and "nuts" for Food-11. (The trigger "flower" was not used in the Food-11 dataset as it had already been employed in the Imagenet-Dogs dataset, and we aimed to explore the effectiveness of diverse triggers.)

Table 8: Embedding Success Rates of different triggers selected by LLM.

| Imagenet-Dogs | | | | | | | | | |
|---|---|---|---|---|---|---|---|---|---|
| **Trigger** | harness | water bowl | toy | bone | blanket | ID tag | food bag | red flower | pet carrier | brush |
| **Embedding Success Rate** | 0.57 | 0.00 | 0.38 | 0.10 | 0.35 | 0.00 | 0.00 | 0.65 | 0.00 | 0.19 |

| Food-11 | | | | | | | | | |
|---|---|---|---|---|---|---|---|---|---|
| **Trigger** | herbs | mint | leaf | lemon slice | strawberry | nuts | ceramic bowl | ice cube | flower | napkin |
| **Embedding Success Rate** | 0.05 | 0.00 | 0.29 | 0.00 | 0.60 | 0.49 | 0.05 | 0.00 | 0.71 | 0.48 |

### A.3 ADDITIONAL EXPERIMENT RESULTS

### A.3.1 ATTACK EFFECTIVENESS

Here are experiments on two network architectures and two datasets with a poisoning rate of 10%. As we find in our paper's main body, the clean accuracy (ACC) of our method is high compared to other attacks, and we achieve the same level of ASR. The attack effectiveness experiments indicate that our VSSC trigger is universally effective regardless of the poisoning ratio.

### A.3.2 DISTORTION IN PHYSICAL SPACE

In our main paper, we only show our method's effectiveness in physical space on ResNet18, with a poison ratio of 10%. Here we present other results under two architectures and two poison ratios in Table 16 17 18. Our attack method demonstrates remarkable robustness across varying network architectures and poison ratios in physical environments. It successfully maintains the accuracy of

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

clean images and can effectively mislead poisoned samples to the target label, despite the visual distortions in the physical space.

For our attack method, we adopt the methodology from (Li et al., 2021c) and implement it in a more comprehensive experimental setup. We introduce variations in lighting, distance, and angle during the image recapture process. Some of these photos are presented in Figure 11. The VSSC triggers embedded in the image demonstrate variable brightness, size, and angle relative to the image content, thus presenting significant diversity. Consequently, the model naturally learns these variations during the attack process. As a result, our method displays resistance to changes in lighting, distance, and angle at the inference stage.

Table 16: Attack performance under physical distortion on ResNet18 with poison ratio=5%.

| | ImageNet-Dogs | | | FOOD-11 | | |
|---|---|---|---|---|---|---|
| Attack | ACC | ASR | RA | ACC | ASR | RA |
| BadNets | 0.88 | 0.00 | 0.81 | 0.50 | 0.07 | 0.43 |
| Blended | 0.83 | 0.26 | 0.60 | 0.67 | 0.33 | 0.33 |
| BPP | 0.71 | 0.10 | 0.55 | 0.67 | 0.03 | 0.43 |
| Input-Aware | 0.71 | 0.00 | 0.71 | 0.70 | 0.00 | 0.57 |
| SIG | 0.90 | 0.10 | 0.55 | 0.70 | 0.33 | 0.27 |
| SSBA | 0.88 | 0.33 | 0.55 | 0.57 | 0.77 | 0.13 |
| TrojanNN | 0.69 | 0.07 | 0.69 | 0.60 | 0.20 | 0.50 |
| WaNet | 0.48 | 0.76 | 0.17 | 0.47 | 0.20 | 0.43 |
| VSSC-redflower | 0.93 | 0.98 | 0.02 | - | - | - |
| VSSC-nuts | - | - | - | 0.70 | 0.63 | 0.17 |

Table 17: Attack performance under physical distortion on VGG19-BN with poison ratio=5%.

| | ImageNet-Dogs | | | FOOD-11 | | |
|---|---|---|---|---|---|---|
| Attack | ACC | ASR | RA | ACC | ASR | RA |
| BadNets | 0.86 | 0.00 | 0.05 | 0.77 | 0.00 | 0.77 |
| Blended | 0.88 | 0.48 | 0.79 | 0.73 | 0.20 | 0.43 |
| BPP | 0.81 | 0.00 | 0.43 | 0.63 | 0.00 | 0.73 |
| Input-Aware | 0.9 | 0.00 | 0.71 | 0.67 | 0.03 | 0.53 |
| SIG | 0.83 | 0.05 | 0.81 | 0.77 | 0.20 | 0.43 |
| SSBA | 0.88 | 0.33 | 0.45 | 0.83 | 0.60 | 0.03 |
| TrojanNN | 0.67 | 0.33 | 0.57 | 0.53 | 0.17 | 0.53 |
| WaNet | 0.74 | 0.12 | 0.55 | 0.27 | 0.40 | 0.00 |
| VSSC-redflower | 0.83 | 0.90 | 0.52 | - | - | - |
| VSSC-nuts | - | - | - | 0.70 | 0.67 | 0.13 |

Table 18: Attack performance under physical distortion on VGG19-BN with poison ratio=10%.

| | ImageNet-Dogs | | | FOOD-11 | | |
|---|---|---|---|---|---|---|
| Attack | ACC | ASR | RA | ACC | ASR | RA |
| BadNets | 0.81 | 0.00 | 0.81 | 0.77 | 0.00 | 0.70 |
| Blended | 0.86 | 0.36 | 0.50 | 0.80 | 0.13 | 0.50 |
| BPP | 0.71 | 0.00 | 0.71 | 0.73 | 0.00 | 0.60 |
| Input-Aware | 0.93 | 0.00 | 0.76 | 0.63 | 0.03 | 0.63 |
| SIG | - | - | - | 0.77 | 0.03 | 0.47 |
| SSBA | 0.9 | 0.26 | 0.60 | 0.73 | 0.50 | 0.07 |
| TrojanNN | 0.57 | 0.50 | 0.40 | 0.57 | 0.37 | 0.37 |
| WaNet | 0.74 | 0.05 | 0.57 | 0.47 | 0.23 | 0.03 |
| VSSC-redflower | 0.83 | 0.83 | 0.12 | - | - | - |
| VSSC-nuts | - | - | - | 0.80 | 0.73 | 0.07 |

| Original | Lighting | Distance | Angle |
|---|---|---|---|
| ASR=0.93 | ASR=0.76 | ASR=0.62 | ASR=0.69 |

Figure 11: Some recaptured photos of printed poisoned images under different conditions.

### A.3.3 ABLATION STUDY FOR POISON RATIO

Figure 12 shows how ACC and ASR change with poison ratio. As the poison ratio increases, our attack method can ensure the improvement of ASR while maintaining the stability of ACC. This also reflects that our attack method has almost no impact on the clean ACC.

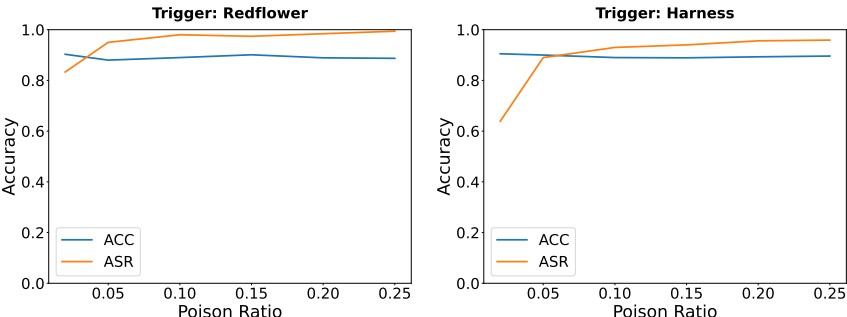

Figure 12: The effect of poisoning rate in digital space, tested on two triggers.