# OpenReview forum: "Robust Backdoor Attack with Visible, Semantic, Sample-specific and Compatible Triggers"
_ICLR.cc/2024/Conference — Submitted to ICLR 2024_

### Official Review · Reviewer_TWWZ · 2023-10-25

**Soundness:** 2 fair
**Presentation:** 3 good
**Contribution:** 1 poor
**Rating:** 3
**Confidence:** 4

**Summary:**

A robust backdoor attack with a visible, semantic, sample-specific, and compatible (VSSC) trigger is proposed. Extensive evaluations have been conducted to show the effectiveness of VSSC and its resistance to various defenses.

**Strengths:**

* The proposed trigger has been evaluated on various datasets and compared with many existing works.

**Weaknesses:**

* The depth of the work is insufficient and the contribution is limited.

The VSSC trigger proposed in the paper is essentially a 'physical' trigger generated digitally. Similar attacks have been proposed in one of the seminal works of backdoor [1]. It is a well-known fact that poisoning the training data with only tens of images of a face wearing a pair of sunglasses can lead to a high attack success rate. Note that the poisoning in [1] uses a weak blended trigger -- directly using real photos for poisoning should require an even lower poisoning ratio.

Besides, the proposed method for the generation of the trigger seems unnecessary in practice (with other concerns in the sequel). Training a generative model is costly while using open-sourced models will limit the application domain (based on the ASR vs PR curves in Figure 6).

Unfortunately, this work has limited contribution due to a lack of additional intellectual merit.

[1] Chen et al, Targeted backdoor attacks on deep learning systems using data poisoning, 2017.

* The methodology needs more consideration.

The reasoning behind the trigger assessment method lacks clarity. There is no evidence showing that models will outperform humans in recognizing the injected trigger objects. It is also unclear why the trigger will be learned if it is recognizable by an object detector. The entire pipeline appears to be heuristic.

* The performance gain of the proposed method over existing ones is marginal.

Also, the poisoning ratio of the proposed trigger is very large.

* Omission of existing works.

The generative model in the proposed work is not state-of-the-art (see [2]). Visible sample-specific triggers have been studied by an early defense in [3].

[2] Liu et al, Cones: Concept Neurons in Diffusion Models for Customized Generation, 2023.
[3] Xiang et al, Revealing perceptible backdoors in DNNs, without the training set, via the maximum achievable misclassification fraction statistic, 2020.

**Questions:**

See the weakness.

---

> ### Author Response · Authors · 2023-11-19
> **Response to Reviewer TWWZ (Part 1)**
>
> Dear Reviewer TWWZ,
>
> We sincerely appreciate your precious time and constructive comments, and are greatly encouraged by your high recognition of the adequacy of our experiments. In the following, we would like to answer your concerns separately.
>
> ---
>
> **Q1: Similar attacks have been proposed in [1], and they used less poisoned images and a weak blended trigger.**
>
> **R1**: As mentioned by reviewer aqPM, **our approach can be seen as an evolved version of [1]**, with the following key distinctions:
>
> 1. **Method to obtain poisoned images.** We would like to clarify that according to the experimental settings of physical attack in [1], in addition to blended triggers, **manually taken photos also need to be used**. This is in line with the limitations of physical attacks we summarized in the common response, that is, **manually taken photos are required**.
> 2. **Trigger selection range.** [1] only uses **simple facial decorations** (glasses) as triggers, and the triggers in both the training and inference stages are the same objects. This is a more ideal and simple scenario as we mentioned in our common response.
> 3. **Trigger selection criteria.** The triggers in [1] are directly specified by the authors **without a scientific basis for selection**, which aligns with our summary of the limitations of physical attacks in the common response.
> 4. **Trigger size.** In [1], a **large trigger size** is required to successfully attack. The sunglasses cover almost 1/4 of the image, resulting in better effectiveness, while read glasses are smaller and less effective in attack.
>
>
>
> For your convenience, we also summarize the difference between us in **Table 1**.
>
> **Table 1: The differences between Blended[1] and VSSC.**
> |   |**Blended[1]**|**VSSC (Ours)**|
> |---|---|---|
> |**Trigger Selection Range**|Limited to simple facial decorations (glasses)|Broad, determined by generative model's capabilities|
> |**Trigger Size**|Large (covers about 1/4 of the image)|Automatically generated, flexible.|
> |**Trigger Selection Criteria**|No criteria, manually specified|LLM-based automatic selection|
> |**Method to Obtain Poisoned Images**|Requires manually taken photos|Automatically generated by generative model in digital space|
>
> It is evident that our method **fundamentally differs from** and **offers multiple advantages over** [1], that’s why reviewer aqPM considers us to be **an evolved version of prior physical attacks**, such as the one outlined in [1].
>
>
> [1] Chen et al, Targeted backdoor attacks on deep learning systems using data poisoning, 2017.
>
> ---
>
> **Q2: Training a generative model is costly while using open-sourced models will limit the application domain.**
>
> **R2**: We believe that this is a misunderstanding of the usage of the generative model in our method, and actually it won't limit the application domain of our attack. We will clarify from the following three aspects:
>
>
> - **We don't need to train a generative model for our attack task.** The task we need to accomplish with generative models is text-guided image editing, for which there are already mature technologies available. Given that current generative models are capable of various tasks such as image generation and editing, they **fully meet our requirements**. As demonstrated in Section 4.2, we use a stable diffusion model (v1.5) pre-trained on general images, which **already has sufficient capacity to meet our needs**, eliminating the need for additional fine-tuning.
> - **The time for preparing the poisoned samples in data poisoning attack is not a barrier.** In data poisoning attacks, poisoned samples are prepared beforehand and there is no direct confrontation between the attacker and the defender. This means that **the effectiveness of the attack is not tied to the time required to prepare poisoned samples**.
> - **We consume much less manpower than traditional physical attacks.** If using traditional physical attacks to achieve similar effects to our attack, it would require manually taking photos, which **consumes more manpower and time**. Our approach is less costly and more convenient in comparison.
>
> In summary, the utilization of generative models is not a limitation for us; on the contrary, by leveraging existing pre-trained generative models, we have opened a pathway for the rapid development of physical attacks. This approach **overcomes the drawback of requiring extensive manpower** associated with physical attacks and will significantly boost the development of physical attacks.

---

> ### Author Response · Authors · 2023-11-19
> **Response to Reviewer TWWZ (Part 2)**
>
> **Q3: The reasoning behind the trigger assessment method lacks clarity. The entire pipeline appears to be heuristic.**
>
> **R3**: We would like to further explain the rationale of the Quality Assessment Module:
>
> - **Necessity of the Quality Assessment Module.** As demonstrated in Section 4.2, we observed that sometimes the generative model may not fulfill our VSSC requirements in a single generation. Therefore, some quality control is essential.
> - **Why we choose dense caption**. The dense caption method can model the relationship between the trigger and surrounding objects. This description helps determine if the generated trigger meets our requirements. Additionally, the triggers we use may possess specific attributes (like a flower with red color), which can be identified by dense captioning.
> - **Why not introduce humans in the loop.** One of the goals of our trigger generation pipeline is to liberate manpower, aligning with our initial intention. Our Human Inspection Study in Section 6 also indicates that images not conforming to human judgment are already eliminated in this process, resulting in high-quality, indistinguishable poisoned samples, **achieving the effect of human inclusion but with labor savings**.
>
> From the above three points, we can see that the design of our Quality Assessment Module is well-motivated. More generally, the logic behind **the design of the entire poisoned dataset generation stage** is distinctly clear and is demonstrated as follows:
> 1. **We start with an analysis of the shortcomings of existing backdoor attacks.** In digital attacks, triggers are prone to failure under visual distortions and are challenging to adapt to physical spaces. In physical attacks, obtaining poisoned samples is highly labor-intensive, and there are limitations such as a narrow range of trigger selections.
> 2. **These shortcomings inspire us to propose four essential characteristics for an ideal trigger: Visible, Semantic, Sample-Specific, and Compatible**. These characteristics ensure the trigger's robustness to visual distortion, its generalizability to the real world, and its stealthiness.
> 3. **To achieve these goals, we designed a poisoned dataset generation pipeline.** Every part of our pipeline is designed around these characteristics: the use of LLM ensures the trigger's compatibility while expanding the selection range of triggers; generative models fulfill visibility, semantic, and sample-specific needs, and the Quality Assessment Module further ensures semantic and compatibility.
>
> In summary, our entire method displays clear motivation, explicit goals, and careful design. **Our motivation, the required characteristics of the trigger, and the corresponding implementation methods are consistent and aligned**, not heuristics.
>
> ---
>
> **Q4: The performance gain of the proposed method over existing ones is marginal.**
> **R4**: As highlighted in the common response, a significant distinction between our attack method and existing ones is its applicability across multiple scenarios. In contrast, most existing attacks are not adaptable across these varied scenarios. We evaluate the performance of our attack method compared to others in the following aspects:
>
> - **Effectiveness in Digital Scenario:**
>     - **Without Visual Distortions.** In the ideal scenario in digital space, our attack method achieves competitive results both with and without backdoor defenses.
>     - **With Visual Distortions.** When faced with visual distortions in digital space, such as blur, noise, and compression, **no other attack method still works** under these distortions, while **our method remains effective**.
> - **Effectiveness in Physical Scenario:**
>     - **With Distortions from Recapturing and Scanning.** In scenarios with visual distortions in physical space, like changes in lighting or clarity due to recapture and scanning, **all the baseline methods fail**. However, **our method maintains ideal ACC and ASR**.
>     - **With distortions from Direct Capturing in OOD Physical Environment.** Most stealthy digital attacks face challenges in extending to physical spaces. Traditional physical attacks are **labor-intensive** and have **a limited range of trigger selection**. Our attack method can generate triggers in digital space and make them effective in physical space. It **automates the selection and addition of triggers, saving manpower, providing a basis for trigger selection, and expanding the range** of trigger selection in physical attacks.
>
>
> In summary, our attack method is **versatile**, achieving competitive or significantly superior performance across **multiple scenarios**, which fully demonstrates its superiority.

---

> ### Author Response · Authors · 2023-11-19
> **Response to Reviewer TWWZ (Part 3)**
>
> **Q5: The poisoning ratio of the VSSC trigger is very large.**
> **R5**: We will clarify this issue from the following two aspects:
>
> - **In our main experiments, the poison ratios used were 5% and 10%.** These ratios are not excessively large. The vast majority of digital attacks in the main experiments or ablation studies cover these two proportions. For example, ISSBA[4] and LC[5] both use 10% poison ratio in their main experiments.
> - **In our ablation study related to OOD scenarios, we used a poison ratio ranging from 5% to 30%.** This is a common range in ablation studies of other attack methods. In OOD scenarios, triggers face more environmental variations, so it is necessary to increase the diversity of poisoned samples to achieve stronger generalization capabilities. Hence, the required poison ratio is slightly higher than in standard conditions.
>
> [4] Li, Yuezun, et al. "Invisible backdoor attack with sample-specific triggers."  2021
> [5] Turner, Alexander, Dimitris Tsipras, and Aleksander Madry. "Label-consistent backdoor attacks." 2019
>
> ---
>
> **Q6: The generative model used is not state-of-the-art.**
> **R6**: Thanks for your suggestion. We would like to offer the following clarifications about our method:
>
> - **The image editing technology we currently use can fully meet our requirements.** The task we need to accomplish with generative models is text-guided image editing, for which there are already mature technologies available. The image editing technique we currently use can completely meet our need. We are grateful for the suggestion and will consider experimenting with more advanced technologies to potentially enhance our attack's effectiveness.
> - **The generative model we used is independent of the overall framework**. As mentioned in our paper, VSSC introduces a concept and framework for generating triggers, instead of a specific image editing method. This independence allows for the integration of **evolving generative technologies into our framework**, continuously enhancing the effectiveness of our attacks.
>
> ---
>
> **Q7: Visible sample-specific triggers have been studied by an early defense in [3].**
> **R7**: Thank you for bringing this work to our attention, published in the International Workshop on Machine Learning for Signal Processing, which we had not previously noticed. Our method has fundamental differences from what is mentioned in [3]:
> - **Trigger selection and injection method.** Our process for trigger selection and injection is **automated and based on a systematic approach**, whereas [3] relies entirely on manual effort. As they mentioned, they "spent laborious human effort to choose the object used as the perceptible backdoor pattern and to embed it into the training images to make them seemingly innocuous to humans."
> - **Trigger diversity.** [3] simply pastes the same object to different positions of different images, which is **not strictly sample-specific**. In contrast, our method uses generative models to insert triggers that are compatible with every original image. The added triggers have **varied appearances**, boosting their robustness to distortions and their adaptability in OOD scenarios.
> - **Applicable scenarios.** [3] does not mention the possibility of extending their method to the **physical world**, whereas our VSSC trigger can achieve this.
>
> For your convenience, we also summarize the difference between us in **Table 2**.
>
> **Table 2: The differences between the attack method in [3] and VSSC.**
> |        | **[3]**        | **VSSC (Ours)**       |
> | ----- | ---------- | ----- |
> | **Trigger Selection**    | No criteria, manually specified                                       | LLM-based automatic selection                     |
> | **Trigger Injection**    | Manual pasting, labor-intensive                                       | Automatically generated, flexible                 |
> | **Trigger diversity**    | Different images will use the same trigger, not truly sample-specific | sample-specific, generated based on image content |
> | **Applicable Scenarios** | Only digital space                                                    | Digital space and physical space                  |
>
> As a versatile attack method, The VSSC **fundamentally differs from** and **offers multiple advantages over** [3].
>
> [3] Xiang et al, Revealing perceptible backdoors in DNNs, without the training set, via the maximum achievable misclassification fraction statistic, 2020.
>
> ---
>
> Hope these responses help to address your concerns. And, we are willing to discuss with you about any further concerns.
>
> Thanks again for your constructive comments and your recognition of our efforts.
>
> Sincerely,
>
> Authors

---

> ### Author Response · Authors · 2023-11-21
> **Awaiting Your Feedback**
>
> Dear Reviewer TWWZ,
>
> We appreciate the insightful and constructive comments you provided on our work.
>
> We've tried our best to carefully consider and address the concerns you raised during the rebuttal process. Your feedback is of great importance in improving the quality of our work.
>
> We understand the constraints of your schedule and highly value the contribution you have made to our work. As the rebuttal period is nearing its end, we are eager to know if our response solves all your questions, and looking forward to receiving your valuable feedback.
>
> Thank you again for your expertise and time.
>
> Sincerely,
>
> Authors

---

> ### Comment · Reviewer_TWWZ · 2023-11-23
> **Response to authors**
>
> I appreciate the extensive efforts made by the authors and their detailed responses to each of my comments. However, I am still concerned about the limitations in the contribution of this work.
>
> The advantage of the proposed VSSC over the attack in [1] may be overstated:
> * It is unclear to me why the attack in [1] is limited to triggers based on facial decorations. This kind of trigger can certainly be used for other domains by its design.
> * Why does the trigger have to be large? For example, a very small yellow sticker on a stop sign can be an effective trigger.
> * The proposed attack assumes the attacker has full control of the input to inject a trigger. But a real physical attack can be achieved by just placing a trigger object in the scene. Besides, taking a few photos is not costly in many cases while digital images are also created by by taking photos.
>
> My comment 'Training a generative model is costly while using open-sourced models will limit the application domain.' is not properly addressed. I don't think open-sourced stable diffusion models can generate a trigger 'tumor' that looks natural to medical images based on my past experience. Similarly, stable diffusion models cannot handle HSI. In these cases, the attacker certainly needs to train a specific generative model, which is very costly.
>
> I totally understand the authors' motivation for getting rid of human labor. However, the cost of achieving this goal outweighs the gain. I will take the responses into consideration when discussing with other reviewers.

---

> > ### Author Response · Authors · 2023-11-23
> >
> > Dear Reviewer TWWC:
> >
> > Our method offers a lower cost in creating poisoned images and more flexibility in scenarios compared to [1]:
> >
> > - **Our method liberates the human resource limitations inherent in traditional physical attack methods.** Our method does not need to manually capture poisoned images, nor manually set up to simulate changes in lighting, background, and other shooting conditions, saving a significant amount of manpower. Contrary to what reviewer [1] suggested, that ”Taking a few photos is not costly in many cases, while digital images are also created by taking photos.” is completely wrong, since we are discussing a data poisoning attack, only the poisoned images need to be taken by the attacker. The “digital images” do not need to be prepared by the attacker. We need to compare the attacker's cost of attack, not the cost of obtaining all images in the dataset.
> > - **Our method also overcomes the scene limitations of traditional physical attack methods.** Our method can generate poisoned images in the digital space, making it effective even for categories that are difficult to capture manually, like rare animals or unusual scenes, expanding the effective range of traditional physical attacks. The experiments in [1] only demonstrate that choosing two types of glasses can make the attack effective. Moreover, when the trigger changes from sunglasses to smaller reading glasses, there is a significant drop in the attack success rate. So **there is no evidence that this kind of trigger can be used for other domains by its design**. [1] also does not prove that choosing other types of triggers is effective. For some categories in natural images, it is difficult for attackers to find suitable scenes for photography, such as rare animals or unusual scenes. As for the yellow sticker you mentioned, it is unnatural for a yellow sticker to appear on a stop sign, failing to meet the stealthiness requirement.
> > - We don’t need to generate images for all domains using the same generative model, just as physical attacks do not take photos of different domains in the same scene, which is an overly harsh requirement. In fact, for different types of images, it is easy to find corresponding domain-specific pre-trained models. For constructing poisoned images through manual photography, finding subjects to photograph and simulating different environmental conditions are labor-intensive and time-consuming. This is far more complicated than training a model.
> >
> > **All the responses in the section about reference [1] were strictly in accordance with its content and experimental setup**. Compared to our method, [1] is limited in scenarios and requires a significant amount of human effort due to the need for manually photographing poisoned images.
> >
> > Hope this response can answer your questions!
> >
> > [1] Chen et al, Targeted backdoor attacks on deep learning systems using data poisoning, 2017.
> >
> > Sincerely,
> >
> > Authors

---

### Official Review · Reviewer_aqPM · 2023-10-31

**Soundness:** 2 fair
**Presentation:** 3 good
**Contribution:** 2 fair
**Rating:** 5
**Confidence:** 3

**Summary:**

This paper introduces an approach to designing backdoor attacks by  leveraging large models (diffusion model and large language model) that are highly effective and robust against visual distortions.
The authors highlight the limitations of existing backdoor attacks, particularly their susceptibility to visual distortions during inference. To address this issue, they propose the use of VSSC triggers that are both effective and resilient to visual distortions. These triggers are designed to have a significant magnitude, increase detection complexity, have semantic meaning, and blend seamlessly with the image content. The authors propose a novel approach to implement VSSC triggers using large language models and text-guided image editing techniques. Extensive experiments validate the effectiveness, stealthiness, and robustness of the VSSC triggers, showcasing their superiority compared to state-of-the-art backdoor attacks. The paper also highlights the advantages of the proposed method in both digital and physical spaces.

**Strengths:**

1. The paper utilizes a large language model to autonomously determine the objects to be introduced into the images and subsequently employs a diffusion model for inpainting. This approach can be seen as an evolved version of prior physical attacks, such as the one outlined in [1], where the selection of glasses is manually determined by humans.

2. The proposed method is capable of successfully executing an attack with a modest poisoning ratio of 5%.

[1] Chen, Xinyun, et al. "Targeted backdoor attacks on deep learning systems using data poisoning." arXiv preprint arXiv:1712.05526 (2017).

**Weaknesses:**

1. The performance of the proposed method is not very impressive. For instance, in Table 3,4,9, several baseline methods outperform the proposed approach.

2. The analysis of robustness to distortion is somewhat limited, considering only the impact of blurring, compression, and noise. Other physical attack papers, such as [1], take into account additional distortions like transformations, shrinking, and padding. These should be discussed as well.

3. An ablation study discussing the influence of the poisoning ratio should be included.


[1] Wenger, Emily, et al. "Backdoor attacks against deep learning systems in the physical world." Proceedings of the IEEE/CVF Conference on Computer Vision and Pattern Recognition. 2021.

**Questions:**

1. If the trigger objects proposed by the Large Language Model (LLM) belong to another class from the training dataset, it could potentially lead to undesirable outcomes, such as misclassification or confusion in the model's predictions. Should the model avoid letting this happen? How  the model avoid letting this happen?

---

> ### Author Response · Authors · 2023-11-19
> **Response to Reviewer aqPM (Part 1)**
>
> Dear Reviewer aqPM,
>
> We sincerely appreciate your precious time and constructive comments, and are greatly encouraged by your high recognition of our **advancements compared to prior physical attacks** and **competitive attack effectiveness**. In the following, we would like to answer your concerns separately.
>
> ---
>
> **Q1: The performance of the proposed method is not very impressive.**
>
> **R1**: We believe there's a **misunderstanding** of our method by the reviewer. As highlighted in the common response, a significant distinction between our attack method and existing ones is its applicability across multiple scenarios. In contrast, most existing attacks are not adaptable across these varied scenarios. If **evaluated comprehensively from various aspects**, our attack method is very impressive. We evaluate the performance of our attack method compared to others in the following aspects:
>
> - **Effectiveness in Digital Scenario:**
>     - **Without Visual Distortions.** In the ideal scenario in digital space, our attack method achieves competitive results both with and without backdoor defenses.
>     - **With Visual Distortions.** When faced with visual distortions in digital space, such as blur, noise, and compression, **no other attack method still works** under these distortions, while **our method remains effective**.
> - **Effectiveness in Physical Scenario:**
>     - **With Distortions from Recapturing and Scanning.** In scenarios with visual distortions in physical space, like changes in lighting or clarity due to recapture and scanning, **all the baseline methods fail**. However, **our method maintains ideal ACC and ASR**.
>     - **With distortions from Direct Capturing in OOD Physical Environment.** Most stealthy digital attacks face challenges in extending to physical spaces. Traditional physical attacks are **labor-intensive** and have **a limited range of trigger selection**. Our attack method can generate triggers in digital space and make them effective in physical space. It **automates the selection and addition of triggers, saving manpower, providing a basis for trigger selection, and expanding the range** of trigger selection in physical attacks.
>
> In summary, our attack method is **versatile**, achieving competitive or significantly superior performance across **multiple scenarios**, which fully demonstrates its superiority.
>
> ---
>
> **Q2: The analysis of robustness to distortion is limited, additional distortions like transformations, shrinking, and padding in [1] should be discussed as well.**
>
> **R2**: We thank the reviewer for this detailed suggestion. We believe the reviewer has a misunderstanding regarding our experimental settings. We would like to clarify that from the following two aspects:
> - **Our experiments on distortions were indeed following the settings used in Section 5 of [1]**, as mentioned in Section 5.3 of our article. The distortions we employed include not only **blurring, compression, and noise** as discussed in [1], but also **transformations, shrinking, and padding**, which the reviewer mentioned. These additional experiments are placed in Figure 11 in **Appendix A.3.2**.
> - **We also considered more challenging out-of-distribution (OOD) real-world scenarios** in Section 5.4. These experiments further show the robustness of the VSSC trigger in physical environments.
>
> [1] Wenger, Emily, et al. "Backdoor attacks against deep learning systems in the physical world." Proceedings of the IEEE/CVF Conference on Computer Vision and Pattern Recognition. 2021.

---

> ### Author Response · Authors · 2023-11-19
> **Response to Reviewer aqPM (Part 2)**
>
> **Q3: An ablation study discussing the influence of the poisoning ratio should be included.**
>
> **R3**: Thanks for the reviewer's valuable suggestion. In the following, we will clarify the ablation study already presented in our paper and introduce a new ablation study focused on the digital space.
>
> - **Figure 6 in our paper already demonstrates the ablation study using four different triggers on two datasets.** This study focuses on **the most challenging scenarios**, OOD images in digital and physical scenarios. Figure 6 in our paper shows that the ASR increases as the poisoning ratio rises.
> - **We have added an ablation study about the poisoning ratio in digital space in Appendix A.3.3** of our updated version. The updated content is highlighted in blue in the revised version. We also demonstrate these results in **Table 1**.
>
> **Table 1** shows how ACC and ASR change with poison ratio with different triggers in digital space. As the poison ratio increases, our attack method can ensure the improvement of ASR while maintaining the stability of ACC. This also reflects that **our attack method has almost no impact on the clean ACC**, unlike other baseline attacks, we also have advantages in this regard.
>
>
> **Table 1: The effect of poisoning rate in digital space.**
>
> |Poison Ratio|Trigger                      |ACC   |ASR                                          |
> |------------|-----------------------------|------|---------------------------------------------|
> |0.02        |red flower                    |0.90  |0.83                                         |
> |0.05        |red flower                    |0.88  |0.95                                         |
> |0.1         |red flower                    |0.89  |0.98                                         |
> |0.15        |red flower                    |0.90  |0.97                                         |
> |0.2         |red flower                    |0.89  |0.98                                         |
> |0.25        |red flower                    |0.89  |0.99                                         |
> |0.02        |harness                      |0.91  |0.64                                         |
> |0.05        |harness                      |0.90  |0.89                                         |
> |0.1         |harness                      |0.89  |0.93                                         |
> |0.15        |harness                      |0.89  |0.94                                         |
> |0.2         |harness                      |0.89  |0.96                                         |
> |0.25        |harness                      |0.90  |0.96  |
>
> ---
>
> **Q4: How to avoid choosing another class in the dataset as the trigger?**
>
> **R4**: This can be controlled via the prompt. In our process of employing GPT-4 for trigger selection, we indeed included a condition in the prompt to avoid choosing any existing classes from the dataset, such as, "*Do not select any classes in the dataset*”. We've added the details of the trigger selection process in **Appendix A.2.8**.
>
> ---
>
> Hope these responses help to address your concerns. We are willing to discuss with you about any further concerns.
>
> Thanks again for your constructive comments and your recognition of our efforts.
>
> Sincerely,
>
> Authors

---

> ### Author Response · Authors · 2023-11-21
> **Awaiting Your Feedback**
>
> Dear Reviewer aqPM,
>
> We appreciate the insightful and constructive comments you provided on our work.
>
> We've tried our best to carefully consider and address the concerns you raised during the rebuttal process. Your feedback is of great importance in improving the quality of our work.
>
> We understand the constraints of your schedule and highly value the contribution you have made to our work. As the rebuttal period is nearing its end, we are eager to know if our response solves all your questions, and looking forward to receiving your valuable feedback.
>
> Thank you again for your expertise and time.
>
> Sincerely,
>
> Authors

---

### Official Review · Reviewer_a1qc · 2023-11-04

**Soundness:** 2 fair
**Presentation:** 3 good
**Contribution:** 3 good
**Rating:** 5
**Confidence:** 3

**Summary:**

The authors observe that existing backdoor attacks are not robust to visual distortions like Gaussian blurring or changes in environmental conditions, which could hinder their practical application.

To mitigate this issue, the paper introduces the concept of a Visible, Semantic, Sample-Specific, and Compatible (VSSC) trigger. The development of the VSSC trigger leverages large language models to select an appropriate trigger, and harnesses text-guided image editing methods to embed the trigger into the poisoned image seamlessly.

The authors demonstrate the VSSC trigger’s performance through rigorous experimentation, showing that it not only retains the stealth required for a successful backdoor attack but also exhibits an enhanced resistance to visual distortions, surpassing the robustness of most existing digital backdoor attacks. Furthermore, the VSSC trigger offers a more effective and adaptable integration approach compared to traditional physical backdoor attacks.

**Strengths:**

- I think the proposed attack is interesting and inspiring. It might makes backdoor detection more challenging.
- The presentation is motivating and easy to follow.

**Weaknesses:**

- The first 5 page writing is great, but the latter evaluation part does not support the claims well. The trigger generation in Design and Evaluation do not match.

**Questions:**

1. In Section 4.2 Stage 1, the paper said LLM are used to automatically select text trigger. But this part is not mentioned in the evaluation. Could the author explain it in detail? How are the evaluated triggers selected?

2. For Table 3 and Table 4, I am not clear why the lowest ACC is highlighted. Isn't that an effective backdoor attack should have a high ACC (and a high ASR at the same time)?

3. Also, from Table 3 and Table 4, I am not convinced that the proposed method is better than baselines. Many bold values are from baselines. This does not align with corresponding text explanation in the paper. Could the author explain it in detail?

4. Minor. In Section 4.2, the algorithm will be more clear and precise if the author can use a pseudo-code, rather than natural language. Also, some adopted techniques are just a reference (like image editing and dense caption). It would be better to provide brief description for better reading experience.

---

> ### Author Response · Authors · 2023-11-19
> **Response to Reviewer a1qc (Part 1)**
>
> Dear Reviewer aiqc,
>
> We sincerely appreciate your precious time and constructive comments, and and are greatly encouraged by your consideration of our work as **interesting and inspiring**. In the following, we would like to answer your concerns separately.
>
> ---
>
> **Q1: How to select candidate triggers using LLM?**
>
> **R1**: Due to space limitations, we limited the display of this part in previous versions, and we've added this part to **Appendix A.2.8**. The updated content is highlighted in blue in the revised version. The specific process is as follows:
>
> 1. **Obtain a candidate trigger list from a large language model (GPT-4) using a pre-designed prompt**. Here's an example of the prompt we used: "*I have a dataset of images from various categories ([List classes in the dataset]), and I wish to add some common but unobtrusive items or decorations to these images. These items should be adaptable to multiple scenarios, harmonious with the theme, and appear natural in at least 80% of the categories in my list. They should be real, concrete objects and not include humans or any classes in the dataset. What items would you recommend?*"
> 2. **Randomly select some images to form a trigger evaluation set.** In our experiments, we randomly selected 10 images from each class in the dataset to constitute the evaluation set.
> 3. **Apply the Trigger Insertion Module to the evaluation set.**
> 4. **Use the Quality Assessment Module to calculate the injection success rate of each candidate trigger.** The success rates of inserting different triggers for the ImageNet-Dogs and FOOD-11 datasets are shown in **Table 1** and **Table 2**.
> 5. **The trigger with the highest success rate is selected as the text trigger.** For example, we use "red flower" and "harness" as triggers for  Imagenet-Dogs, and "strawberry" and "nuts" for Food-11.
>
>
> **Table 1: Embedding Success Rate (ESR) of Candidate Triggers of Imagenet-Dogs Dataset**
>
> | Trigger | harness                                         | water bowl | toy    | bone        | blanket     | ID tag | food bag     | red flower | pet carrier | brush       |
> |---------------|--------------------|------------|--------|-------------|-------------|--------|--------------|------------|-------------|-------------|
> |**ESR** | 0.57      | 0.00          | 0.38  | 0.10 | 0.35 | 0.00      | 0.00            | 0.65 | 0.00           | 0.19 |
>
>
> **Table 2: Embedding Success Rate (ESR) of Candidate Triggers of FOOD-11 Dataset**
>
> | Trigger | herbs       | mint | leaf   | lemon slice | strawberry  | nuts  | ceramic bowl | ice cube | flower  | napkin |
> |-----------|-----------|------|--------|-------------|-------------|-------|--------------|----------|---------|--------|
> |**ESR** | 0.05 | 0.00    | 0.29 | 0.00  | 0.60 | 0.49 | 0.05       | 0.00        | 0.71 | 0.48  |
>
> (The trigger "flower" was not used in the Food-11 dataset as it had already been employed in the Imagenet-Dogs dataset, and we aimed to explore the effectiveness of diverse triggers.)
>
> ---
>
> **Q2: Why the lowest ACC is highlighted in Table 3 and Table 4?**
>
> **R2**: Figures 3 and 4 demonstrate the resistance abilities of different attacks when facing defenses. The objective of a defense is to reduce the ASR while keeping the ACC. Therefore, a lower ACC indicates poorer defense effectiveness against that particular attack method, which also means better resistance performance of the attack method against the defense. This is why we have highlighted the lowest ACC values. Thanks for this concern, to avoid misunderstanding, we've clarified this in **Section 5.1 of our revised manuscript and highlighted it in blue**.

---

> ### Author Response · Authors · 2023-11-19
> **Response to Reviewer a1qc (Part 2)**
>
> **Q3: The results in Tables 3 and 4 are not convincing to the reviewer that the VSSC trigger is better than baseline methods.**
>
> **R3**: As highlighted in the common response, a significant distinction between our attack and existing ones is its applicability across multiple scenarios. In contrast, most existing attacks are not adaptable across these varied scenarios. If **evaluated comprehensively from various aspects**, our attack method is better than baseline methods in most aspects. We evaluate the performance of our attack method compared to others in the following aspects:
>
> - **Effectiveness in Digital Scenario:**
>     - **Without Visual Distortions.** In the ideal scenario in digital space, our attack method achieves competitive results both with and without backdoor defenses.
>     - **With Visual Distortions.** When faced with visual distortions in digital space, such as blur, noise, and compression, **no other attack method still works** under these distortions, while **our method remains effective**.
> - **Effectiveness in Physical Scenario:**
>     - **With Distortions from Recapturing and Scanning.** In scenarios with visual distortions in physical space, like changes in lighting or clarity due to recapture and scanning, **all the baseline methods fail**. However, **our method maintains ideal ACC and ASR**.
>     - **With distortions from Direct Capturing in OOD Physical Environment.** Most stealthy digital attacks face challenges in extending to physical spaces. Traditional physical attacks are **labor-intensive** and have **a limited range of trigger selection**. Our attack method can generate triggers in digital space and make them effective in physical space. It **automates the selection and addition of triggers, saving manpower, providing a basis for trigger selection, and expanding the range** of trigger selection in physical attacks.
>
> In summary, our attack method is **versatile**, achieving competitive or significantly superior performance across **multiple scenarios**, which fully demonstrates its superiority.
>
> ---
>
> **Q4: Use a pseudo-code in Section 4.2, and provide a brief description for image editing and dense caption techniques.**
>
> **R4**: Thanks for your constructive suggestions, we have added a brief description of these two technologies to the article, and added the pseudo-code of the pipeline in Section 4.2 to **Appendix A.1**. The updated content is highlighted in blue in the revised version. Also, the image editing method and trigger embedding evaluation method we used are independent of the overall framework. As mentioned in our paper, VSSC introduces a concept and framework for generating triggers, instead of a specific image editing method or trigger embedding evaluation method. This independence allows for the integration of evolving generative technologies into our framework, continuously enhancing the effectiveness of our attacks.
>
> ---
>
> Hope these responses help to address your concerns. And, we are willing to discuss with you about any further concerns.
>
> Thanks again for your constructive comments and your recognition of our efforts.
>
> Sincerely,
>
> Authors

---

> ### Author Response · Authors · 2023-11-21
> **Awaiting Your Feedback**
>
> Dear Reviewer a1qc,
>
> We appreciate the insightful and constructive comments you provided on our work.
>
> We've tried our best to carefully consider and address the concerns you raised during the rebuttal process. Your feedback is of great importance in improving the quality of our work.
>
> We understand the constraints of your schedule and highly value the contribution you have made to our work. As the rebuttal period is nearing its end, we are eager to know if our response solves all your questions, and looking forward to receiving your valuable feedback.
>
> Thank you again for your expertise and time.
>
> Sincerely,
>
> Authors

---

> > ### Comment · Reviewer_a1qc · 2023-11-23
> >
> > Thank the authors for their efforts in the rebuttal. However, I am not convinced and I keep the original score.

---

> > > ### Author Response · Authors · 2023-11-23
> > > **Response to Reviewer a1qc**
> > >
> > > Dear Reviewer a1qc,
> > >
> > > Thank you for your consideration of our rebuttal. Could you please specify which aspects of our rebuttal you found unconvincing? Any additional insights you could provide would be extremely valuable to us in improving our manuscript and addressing your concerns more thoroughly.
> > >
> > > Sincerely,
> > >
> > > Authors

---

### Author Response · Authors · 2023-11-19
**A common response to questions about the contribution of VSSC (Part 2)**

**Advantages of Our VSSC Trigger**

The primary contribution of the VSSC trigger lies in its stability when facing visual distortions, and its capability to generalize to the real world. It not only **addresses the inherent limitations in traditional digital and physical attacks** but also achieves competitive results under and without defenses in digital space.

- **Compared to Digital Attacks**
    - **Jump Out of the Trap of Stealthiness and Robustness Dillemma:** Through its semantic and compatibility features, the VSSC trigger maintains stealthiness while making sufficient alterations to images, thus ensuring stability against visual distortions from image processing.
    - **Extension to the Physical World:** Triggers generated via diffusion **inherently vary in size, angle, and brightness, simulating different physical environmental changes**. Consequently, the model learns to recognize various physical distortions during training, maintaining trigger effectiveness even after recapturing or scanning. Additionally, owing to its visible and semantic characteristics, **the VSSC trigger can easily find corresponding objects in the real world** to serve as triggers.

- **Compared to Physical Attacks**

     As mentioned by reviewers aqPM, our VSSC trigger is an evolved version of prior traditional physical attacks.
    - **Automated Trigger Addition**: An efficient pipeline has been designed for the automated selection and addition of triggers. The introduction of generative models allows poisoned images to be generated entirely in the digital space, eliminating the need for manual photography or editing, thus significantly reducing the costs of physical attacks.
    - **Easily Enhancing Robustness:** The use of the diffusion model introduces diversity, removing the need for manually capturing images under various environmental conditions. Experiments on OOD data demonstrate that the VSSC trigger can adapt to different physical environments.
    - **Systematic Trigger Selection Using a Large Language Model:** Our pipeline leverages LLM's prior knowledge, providing a basis for trigger selection. This makes the trigger selection process more systematic, avoids the arbitrariness of artificially designated triggers, and expands the range of trigger choices.
    - **Trigger Appearance Flexibility:** VSSC only needs triggers in training and inference stages that share the **same semantics**, not necessarily the same object.


Our method has been recognized by reviewer aqPM as an "**evolved version of prior physical attacks**" because it **breaks through the current barriers** hindering the development of traditional physical attacks.

Finally, we want to emphasize the contributions of our work to the field:

1. The concept of VSSC we propose points out **a new direction for effective physical attacks**.
2. Our framework effectively **resolves the constraints of restricted trigger diversity and extensive manpower requirements** in previous physical attacks, which will significantly propel the rapid development of physical backdoor attacks.

The concepts and methods we mentioned will provide a lot of inspiration to future researchers, promoting the advancement of this field.

---

### Author Response · Authors · 2023-11-19
**A common response to questions about the contribution of VSSC (Part 1)**

As **a versatile attack method applicable to both digital and physical scenarios**, the value of our method should be analyzed from multiple aspects.  We will demonstrate it through the following three sections: the first two sections are **the limitations of traditional digital and physical attacks**, and the third section highlights **the advantages of our method** over these existing attacks.


**Limitations of Traditional Digital Attacks**

- **Stealthiness vs. Robustness Dilemma in Digital Space:** Existing digital attacks often link stealthiness with invisibility, creating **a conflict between the stealthiness of the trigger and its robustness against visual distortions**. In ideal scenarios, digital attacks achieve high ASR because the trigger features exhibit minimal variation between training and test datasets, without facing any distortions. However, **introducing distortion during the training stage will affect the effectiveness of invisible triggers.** To the best of our knowledge, no existing attack method can simulate visual distortions as realistically as ours during the training process, making them vulnerable in different conditions.
- **Ineffectiveness under Physical Distortions:** Trigger addition occurs entirely in the digital space, without simulating variations in physical space. **The complex lighting conditions and camera shooting settings in physical environments easily render triggers ineffective**. Consequently, digital attacks lose effectiveness when images are recaptured or scanned. This limits their real-world applicability. Traditional digital attacks are also **difficult to deploy in the real world**. Even for visible triggers, it’s difficult to find the same objects in the physical world.

**Limitations of Traditional Physical Attacks**

- **Manually Constructing Poisoned Samples:** Existing physical attacks require manually constructing poisoned images by taking photos, which is time-consuming and labor-intensive.
- **High Cost to Achieving Robustness:** It is challenging to simulate changes in lighting, distance, and angles in physical environments during attacks. To adapt to these changes, a larger volume of photos must be manually taken under controlled conditions.
- **Absence of a Systematic Trigger Selection Process:** Existing physical attacks lack a systematic process for trigger selection and have a very limited range of choices. Triggers are manually specified, using a heuristic method, often choosing facial decorations for face datasets and stickers for natural object datasets.
- **Restricted Trigger Appearance:** Triggers in the training and inference stages must have **identical appearances**.

---

### Author Response · Authors · 2023-11-19
**Summary of all updates**

We are glad to hear that reviewers think our method is **an evolved version of prior physical attacks** (aqPM) and is **interesting and inspiring** (a1qc). We’d like to express our sincere gratitude to the reviewers for their thoughtful and constructive comments. We appreciate the effort they have put into providing us with insightful feedback, which has been invaluable in enhancing the quality of our work.

Following the reviewers' suggestions, we have updated and uploaded the revised manuscript, **marking all changes in blue**. Should our paper be accepted, we will continue to refine it further. The major updates include:

- Adding the pseudo-code of the poisoned dataset generation stage in Section 4.2 (Appendix A.1).
- Added the detailed process of automatic trigger selection in our experiments (Appendix A.2.8).
- Added an ablation study for the poison ratio in the digital space (Appendix A.3.3).

---

### Author Response · Authors · 2023-11-22
**Eagerly awaiting further discussion with reviewers**

Dear Reviewers,

We hope to further discuss our response to ensure it addresses your concerns. We believe we've clarified key points, but are ready to provide more information if needed.

We genuinely hope reviewers could kindly check our response. Thank you for your time.

Best regards,

Authors

---

### Meta-Review · Area_Chair_M81f · 2023-12-12

**Metareview:**

The authors submitted rebuttal, but two of the reviewers were not convinced and no reviewer changed the rating. In the end, no reviewer suggested accepting the paper, so will not be accepted.

**Justification For Why Not Higher Score:**

No reviewer suggests acceptance. The main weaknesses include:
- The advantage of the proposed method compared with [1]. The paper makes some assumptions about [1], e.g., trigger size, with which the reviewer does not agree.
- the cost of training a generative model
- and limited contribution in general as described in the reviewers.

**Justification For Why Not Lower Score:**

N/A

---

### Decision · Program_Chairs · 2024-01-16

Reject